# Theoretical Perspectives on Data Quality and Synergistic Effects in Pre- and Post-Training Reasoning Models

Adel Javanmard [1 2]   Baharan Mirzasoleiman [3 2]   Vahab Mirrokni [2]

## Abstract

Large Language Models (LLMs) are pretrained on massive datasets and later instruction-tuned via supervised fine-tuning (SFT) or reinforcement learning (RL). Best practices emphasize large, diverse pretraining data, whereas post-training operates differently: SFT relies on smaller, high-quality datasets, while RL benefits more from scale, with larger amounts of feedback often outweighing label quality. Yet it remains unclear why pretraining and RL require large datasets, why SFT excels on smaller ones, and what defines high-quality SFT data. In this work, we theoretically analyze transformers trained on an in-context weight prediction task for linear regression. Our analysis reveals several key findings: $(i)$ balanced pretraining data can induce latent capabilities later activated during post-training, and $(ii)$ SFT learns best from a small set of examples challenging for the pretrained model, while excessively large SFT datasets may dilute informative pretraining signals. In contrast, RL is most effective on large-scale data that is not overly difficult for the pretrained model. We validate these theoretical insights with experiments on large nonlinear transformer architectures.

## 1 Introduction

Pretraining on massive language datasets, followed by post-training, is essential for unlocking and shaping the capabilities of large language models (LLMs). While pretraining endows models with broad linguistic knowledge and general world understanding, post-training transforms these latent capabilities into usable skills that can be reliably elicited through instructions. This transformation is typically achieved through either supervised fine-tuning (SFT), which trains models to imitate high-quality demonstrations, or reinforcement learning (RL), which optimizes model behavior using scalar feedback to refine global properties such as reasoning quality and preference alignment. Despite their central role, the interaction between pretraining data and post-training data—and how this interaction determines the resulting model capabilities—remains poorly understood.

In practice, pretraining commonly relies on massive and diverse data mixtures, whereas post-training follows a variety of recipes. For example, OpenAI o1 (OpenAI, 2024) and DeepSeek R1 (Guo et al., 2025) achieve state-of-the-art reasoning performance through RL applied to large-scale datasets, while s1 (Muennighoff et al., 2025) demonstrates comparable math reasoning performance using SFT on a small, manually curated set of hard and diverse examples. More recently, Llama 4 (Meta, 2025) adopts iterative rounds of SFT and RL on progressively harder data. Yet, which characteristics of pretraining data unlock superior post-training performance, and what requirements on the quality and scale of post-training data are needed to bring a pretrained model to optimal performance, have remained unclear.

In this work, we answer the above questions by studying an in-context weight prediction task for linear regression, where the goal is to predict the linear weight vector from the sequence of input prompts. This framework has been used previously for analyzing the mechanism underlying training CoT (Huang et al., 2025b; Javanmard et al., 2025). In this work, we propose a novel pipeline where during pretraining, the model performs direct in-context-learning and outputs its prediction of the weight vector. During post-training, the transformer performs CoT with SFT or RL and generates multiple intermediate steps before arriving at its final prediction of the weight vector. We test the model on a combination of pretraining and post-training tasks.

While our theoretical setup captures the key distinction between **outcome supervision** (RL, rewarding final answers) and **process supervision** (SFT, supervising intermediate steps), it significantly abstracts from standard RL algorithms

[1]Department of Data Sciences and Operations, Marshall School of Business, University of Southern California, Los Angeles, USA [2]Google Research, New York, USA [3]Department of Computer Science, University of California, Los Angeles, USA. Correspondence to: Adel Javanmard <ajavanma@usc.edu>.

*Proceedings of the $43^{rd}$ International Conference on Machine Learning*, Seoul, South Korea. PMLR 306, 2026. Copyright 2026 by the author(s).

that involve sampling, advantage estimation, and policy gradients. Here, we model RL as outcome-supervised regression on the transformer's in-context prediction task. This simplification enables clean theoretical analysis but limits direct applicability to full RLHF implementations in LLMs.

Our analysis shed light on several questions:

(i) What characteristics of pretraining data enable models to develop latent capabilities that can be effectively unlocked during post-training?

(ii) Given a pretrained model, what properties define effective SFT data that promote adaptation to new skills, while minimizing interference with capabilities acquired during pretraining?

(iii) Given a pretrained model, what properties of RL data are most critical? How does the RL optimization landscape differ from that of SFT, and when can RL achieve outcomes comparable to SFT?

Our analysis helps to rigorously understand several empirically observed phenomena reported in the literature. Specifically, for our in-context setting, it shows that (i) effective pretraining data contains a balanced mixture of data from all categories. Such data can induce latent capabilities that are activated during post-training. (ii) Post-training with SFT benefits the most from a small set of challenging examples for the pretrained model, and larger SFT data can harm the performance. (iii) RL requires large-scale data that is informative but not overly difficult for the pretrained model.

We confirm our findings with experiments on an in-context weight prediction task for linear regression on transformer with a single linear self-attention (LSA), as well as large, nonlinear transformer architectures, namely GPT2 (Radford et al., 2019).

## 2    Related Work

Recent work has highlighted several phenomena relevant to our study.

**Pretraining.** For pretraining LLMs, common practice is to use a large mixture of language data. Recent studies mostly focused on data filtering (Li et al., 2024), data selection (Nguyen et al., 2024; Yang et al., 2024), and mixture reweighting (Xie et al., 2023). Empirically, high-quality pretraining data should be large and diverse. Such high-quality pretraining data can induce latent capabilities that are not necessarily observed after pretraining but are activated during post-training (Akter et al., 2025).

**Post-training.** For post-training, recent studies mostly focused on comparing post-training with SFT and RL (Xiong et al.; Zhao et al., 2025; Aminian et al., 2025). Theoretically, SFT is mode covering: by minimizing forward KL

to demonstration data, it encourages the model to assign probability mass to all plausible responses. In contrast, reinforcement learning (RL) is mode seeking: by optimizing reward (typically under a KL constraint), it concentrates probability on high-reward responses and suppresses lower-ranked alternatives. As a result, SFT defines the space of acceptable behaviors, while RL selects and amplifies the most preferred ones within that space. Empirically, SFT data should be small and high-quality, i.e. hard and diverse (Muennighoff et al., 2025; Guha et al., 2025; Huang et al., 2025b), and larger SFT data washes away benefits of high-quality pretraining data (Akter et al., 2025). In contrast, RL benefits from larger data that is still challenging but not overly difficult for the pretrained model (Zeng et al., 2025; Yue et al., 2025; Meta, 2025).

Nevertheless, the reasons why certain characteristics of pretraining data unlock superior post-training performance, why SFT benefits from a small set of hard and diverse examples while larger datasets can degrade its effectiveness, and why data scale matters more than apparent quality in RL have remained unclear. Our theoretical framework demystifies these observations, bridging the gap between empirical results and a principled understanding of data dynamics.

## 3    Problem Setup

We focus on in-context learning (ICL) setting, where a model is presented with a context dataset $D = \{(x_i, y_i)\}_{i=1}^n$ and each $(x_i, y_i)$ pair is sampled independently from some underlying distribution $P$. Here, the input vectors $\{x_i\}_{i=1}^n$ belong to $\mathbb{R}^d$, and the corresponding labels $\{y_i\}_{i=1}^n$ may be real numbers (for regression tasks) or binary values such as $\{0, 1\}$ (for classification tasks). The model is then given a new test input $x_{n+1} \sim P_x$ and is tasked to predict its associated label or corresponding in-context weight predictor. In other words, in-context learning operates on sequences, called prompts, of input-output pairs $(x_1, y_1, \ldots, x_n, y_n, x_{n+1})$ and each prompt may have its own distribution.

**Linear Self Attention (lSA)** Let $Z$ be an embedding formed from the prompt (We will discuss the specific construction later). The softmax self-attention module takes as input an embedding matrix and outputs a matrix of the same size,

$$f_{\text{Attn}}(Z; W_K, W_Q, W_V, W_P)$$
$$= Z + W_P W_V Z \cdot \text{softmax}\left(\frac{(W_K Z)^\top W_Q Z}{\lambda}\right)$$

where softmax is applied column-wise. In Linear-Self-Attention (LSA) the softmax nonlinearity is removed. By defining $W := W_K^\top W_Q$, $V = W_P W_V$ and $\theta = (W, V)$ we arrive at:

$$f_{\text{LSA}}(Z; \theta) = Z + VZ \cdot \frac{Z^\top W Z}{\lambda} \qquad (3.1)$$

We will focus on in-context linear predictors. Each prompt is of the form $P_\tau = (x_{\tau,1}, y_{\tau,1}, \ldots, x_{\tau,n}, y_{\tau,n}, x_{\tau,n+1})$, with $y_{\tau,i} = \langle w_\tau, x_{\tau,i} \rangle$, where $w_\tau \sim \mathsf{N}(0, I_d)$.

**Supervised Fine-Tuning and Outcome Supervision.** We begin by describing outcome supervision (OS) training with $k$ steps of chain-of-thought reasoning. As noted in the introduction, this formulation simplifies standard RL—which involves sampling, advantage estimation, and policy gradients—by modeling it as outcome-supervised regression that rewards final answers, while still capturing the core distinction from process-supervised SFT.

Suppose we are given a prompt $P_\tau = (x_{\tau,1}, y_{\tau,1}, \ldots, x_{\tau,n}, y_{\tau,n})$. We construct the embedding

$$\hat{Z}_{\tau,0} = \begin{bmatrix} x_{\tau,1} & \ldots & x_{\tau,n} & 0 \\ y_{\tau,1} & \ldots & y_{\tau,n} & 0 \\ 0 & \ldots & 0 & w_{\tau,0} \\ 0 & \ldots & 0 & 1 \end{bmatrix}, \qquad (3.2)$$

and iteratively define $\hat{Z}_{\tau,i+1} = [\hat{Z}_{\tau,i}, f_{\mathrm{LSA}}(\hat{Z}_{\tau,i})_{[:,-1]}]$. We initialize $w_{\tau,0} = 0_{d \times 1}$ and set $\hat{w}_{\tau,i+1} := f_{\mathrm{LSA}}(\hat{Z}_{\tau,i})_{[d+2:2d+1,-1]}$. This yields

$$\hat{Z}_{\tau,i} = \begin{bmatrix} x_{\tau,1} & \ldots & x_{\tau,n} & 0 & * & \ldots & * \\ y_{\tau,1} & \ldots & y_{\tau,n} & 0 & * & \ldots & * \\ 0 & \ldots & 0 & w_{\tau,0} & \hat{w}_{\tau,1} & \ldots & \hat{w}_{\tau,i} \\ 0 & \ldots & 0 & 1 & 1 & \ldots & 1 \end{bmatrix}, \qquad (3.3)$$

Let $w_\tau^*$ be the ground-truth weight for prompt $P_\tau$, for $\tau \in [B]$. The outcome supervision (OS) loss is

$$\mathcal{L}^{\mathrm{OS}}(V, W) = \frac{1}{2B} \sum_{\tau=1}^{B} \|\hat{w}_{\tau,k} - w_\tau^*\|_{\ell_2}^2, \qquad (3.4)$$

i.e., OS penalizes only the final step of the $k$-step reasoning process.

For Supervised fine-tuning (SFT), we use ground-truth chain-of-thought (CoT) sequences

$$Z_{i,\tau} = \begin{bmatrix} x_1 & \ldots & x_n & 0 & * & \ldots & * \\ y_1 & \ldots & y_n & 0 & * & \ldots & * \\ 0 & \ldots & 0 & w_{0,\tau} & w_{1,\tau} & \ldots & w_{i,\tau} \\ 0 & \ldots & 0 & 1 & 1 & \ldots & 1 \end{bmatrix}, \quad (3.5)$$

where $w_{i,\tau} = (1 - (1-\eta)^i) w_\tau^*$ with $w_{0,\tau} = 0$ provides exponentially converging intermediate targets, with an arbitrary but fixed rate $\eta$. We chose this sequence as an analytically clean proxy for gradient-based optimization. Specifically, In population linear regression with identity covariance, gradient descent updates are given by $w_i = w_{i-1} - \eta(w_{i-1} - w_\tau^*)$, giving $w_{i,\tau} = (1 - 1(1-\eta)^i) w_\tau^*$, identical to our CoT intermediate supervising sequence.

The model is trained to predict the next token $Z_{i+1,\tau}[:,-1] := (0_d, 0, w_{i+1,\tau}, 1)$ given $Z_{i,\tau}$. Over $B$ training prompts, the SFT loss is

$$\mathcal{L}^{\mathrm{SFT}}(V, W) :=$$
$$\frac{1}{2B} \sum_{\tau=1}^{B} \sum_{i=0}^{k} \left\| f_{\mathrm{LSA}}(Z_{i,\tau})_{[:,-1]} - (0, 0, w_{i+1,\tau}, 1) \right\|_{\ell_2}^2.$$

*Remark* 3.1 (RL with Process Reward). While this work focuses on outcome supervision as a simplification for RL, recent studies have also explored process rewards and intermediate supervision in RL-based LLM training (Setlur et al., 2025; Cui et al., 2025). From a mathematical perspective, process rewards can be viewed as a partial-supervision variant of our SFT formulation. In standard SFT, supervision is provided at every step, whereas process reward models typically assign feedback only to selected intermediate reasoning milestones. In practice, such rewards are rarely dense at the token level; instead, annotators often provide supervision at discrete points, such as the end of a sentence or a specific derivation step. To formalize this, let $M \subseteq \{1, \ldots, k\}$ denote the subset of steps that receive process reward. This notation smoothly interpolates between several settings: (i) $M = \{1, \ldots, k\}$, corresponding to full SFT supervision; (ii) $M = \{k\}$, corresponding to outcome supervision; and (iii) $1 < |M| < k$, corresponding to intermediate process reward supervision.

**Pipeline: Pre-training, Post-training, Post-testing.** Our pipeline has three stages distinguished by data covariances: pre-training on $\Sigma_0$, post-testing on $\Sigma = \Sigma_0 + \Delta$ (low-rank $\Delta$), and post-training on a chosen intermediate distribution (discussed later for optimal post-test performance). Inputs $x \in \mathbb{R}^d$ are Gaussian throughout.

Assuming infinite pre-training prompts, population analysis of (Huang et al., 2025a) shows that with proper initialization, the pretrained parameters are given by:

$$\hat{V}_0 = \begin{bmatrix} 0 & 0 & 0 & 0 \\ 0 & 0 & 0 & 0 \\ -\Gamma_0^{-1} & 0 & 0 & 0 \\ 0 & 0 & 0 & 0 \end{bmatrix}, \qquad \hat{W}_0 = \begin{bmatrix} 0 & 0 & I & 0 \\ 0 & 0 & 0 & -1 \\ 0 & 0 & 0 & 0 \\ 0 & 0 & 0 & 0 \end{bmatrix}, \qquad (3.6)$$

where

$$\Gamma_0 := \left(1 + \frac{1}{n}\right) \Sigma_0 + \frac{1}{n} \mathrm{tr}(\Sigma_0) I_d \in \mathbb{R}^{d \times d}, \qquad (3.7)$$

with $n$ the prompt length. Post-training initializes from $(\hat{V}_0, \hat{W}_0)$, and updates the transformer weights by minimizing either the SFT loss or the OS loss.

**Sparsity structure motivated by the population regime.** (Huang et al., 2025a) shows that training with chain-of-thought (paralleling our SFT loss) in the population regime

($B \to \infty$ before $d, n$) preserves sparsity in the weights from initialization (3.6). Specifically, Lemma C.2 in (Huang et al., 2025a) proves that the gradient flow trajectory preserves the following sparsity structure:

$$
V(t) = \begin{bmatrix} 0 & 0 & 0 & 0 \\ 0 & 0 & 0 & 0 \\ V_{31}(t) & 0 & 0 & 0 \\ 0 & 0 & 0 & 0 \end{bmatrix}, \quad W(t) = \begin{bmatrix} 0 & 0 & W_{13}(t) & 0 \\ 0 & 0 & 0 & -1 \\ 0 & 0 & 0 & 0 \\ 0 & 0 & 0 & 0 \end{bmatrix},
$$
(3.8)

where $V_{31}(t), W_{13}(t) \in \mathbb{R}^{d \times d}$ are the parameters at time $t$. While their analysis assumes identity-covariance Gaussians and intermediate weights $w_{i,\tau}$ derived from standard gradient descent the proof of Lemma C.2 in (Huang et al., 2025a) relies only on the symmetry properties of $w_\tau^* \sim \mathsf{N}(0, I)$ and the fact that $w_{i,\tau}$ is an odd function of $w_\tau^*$. Consequently, this structural result extends to our setting of general covariances and supervised sequences. Although our analysis moves beyond the population regime, these insights motivate us to constrain our transformer model to follow similar sparsity pattern. Throughout, we use the shorthands $\widetilde{V}$ and $\widetilde{W}$ to indicate the nonzero blocks of $V$ and $W$.

## 4 Analysis of the SFT loss

Let $S_\tau := \frac{1}{n} \sum_{i=1}^n x_{i,\tau} x_{i,\tau}^\mathsf{T}$ be the empirical features covariance for $\tau \in [B]$. We also define the following matrices:

$$
\Omega := [w_1^*, \dots, w_B^*] \in \mathbb{R}^{d \times B}
$$

$$
\Phi := [S_1 w_1^*, \dots, S_B w_B^*] \in \mathbb{R}^{d \times B}, \quad M := \Phi\Phi^\mathsf{T} \in \mathbb{R}^{d \times d}
$$

The next theorem characterizes the minimizer of the SFT loss that is closest to the initialization $(-\Gamma_0^{-1}, I)$.

**Theorem 4.1.** *Define*

$$
(\widetilde{V}_\lambda, \widetilde{W}_\lambda) =
$$

$$
\underset{(\widetilde{V}, \widetilde{W})}{\arg\min} \; \mathcal{L}^{\mathrm{SFT}}(\widetilde{V}, \widetilde{W}) + \lambda \left\| \widetilde{V} + \Gamma_0^{-1} \right\|_F^2 + \lambda \left\| \widetilde{W} - I \right\|_F^2 .
$$

*We then have* $\lim_{\lambda \to 0^+} (\widetilde{V}_\lambda, \widetilde{W}_\lambda) = (\widetilde{V}_*, \widetilde{W}_*)$, *where*

$$
\widetilde{W}_* = I, \quad \widetilde{V}_* = -\eta \Omega \Phi^\dagger - \Gamma_0^{-1}(I - \Phi\Phi^\dagger) \quad (4.1)
$$

Our next theorem shows that the solution $(\widetilde{V}_*, \widetilde{W}_*)$ can be attained by gradient descent initialized at $(-\Gamma_0^{-1}, I)$, and establishes conditions on the step size for convergence along with its convergence rate.

**Theorem 4.2.** *Fix* $\widetilde{W} = I$. *Consider the sequence of weights* $\{\widetilde{V}_t\}_{t \geq 0}$ *generated by the gradient descent update* $\widetilde{V}_{t+1} = \widetilde{V}_t - \gamma \nabla_{\widetilde{V}} \mathcal{L}^{\mathrm{SFT}}(\widetilde{V}_t, I)$ *with initialization* $\widetilde{V}_0 = -\Gamma_0^{-1}$ *and a constant step size* $0 < \gamma$. *Define* $\rho := 1 - \eta$ *and* $c_k := \sum_{i=0}^k \rho^{2i} < \frac{1}{1 - \rho^2} = \frac{1}{2\eta - \eta^2}$. *If* $\gamma < \frac{2B}{c_k \lambda_{\max}(M)}$, *then the GD updates converges to* $\widetilde{V}_*$ *at the following rate:*

$$
\left\| \widetilde{V}_t - \widetilde{V}_* \right\|_F \leq \alpha^t \left\| \Gamma_0^{-1} + \widetilde{V}_* \right\|_F,
$$

$$
\alpha := \max\left( \left| 1 - \frac{\gamma c_k}{B} \lambda_{\max}(M) \right|, \left| 1 - \frac{\gamma c_k}{B} \lambda_{\min}^+(M) \right| \right)
$$

*where* $\lambda_{\max}(M)$ *and* $\lambda_{\min}^+(M)$ *respectively denote the maximum and the minimum (nonzero) eigenvalues of* $M$. *In particular, setting* $\gamma = \frac{B}{c_k \lambda_{\max}(M)}$, *we obtain*

$$
\left\| \widetilde{V}_t - \widetilde{V}_* \right\|_F \leq \left( 1 - \frac{\lambda_{\min}^+(M)}{\lambda_{\max}(M)} \right)^t \left\| \Gamma_0^{-1} + \widetilde{V}_* \right\|_F
$$

*Remark 4.3.* Note that the loss minimizer $(\widetilde{V}_*, \widetilde{W}_*)$ given by (4.1) depends on $n$ (prompt length) and $B$ (number of prompts), the step size $\eta$ in the supervised weight path, but not on $k$ (length of reasoning paths). However, if we fix the gradient step size $\gamma < \frac{2B(2\eta - \eta^2)}{\lambda_{\max}(M)}$, by Theorem 4.2 larger $k$ implies larger $c_k$ and so faster convergence rate.

It is worth deriving the limit of $\widetilde{V}_*$ in the population regime, where $B \to \infty$, while $n, d$ are kept fixed.

**Proposition 4.4.** *Suppose that the features are generated as* $x_{i,\tau} \sim \mathsf{N}(0, A)$ *for a positive semidefinite matrix* $A \in \mathbb{R}^{d \times d}$. *Suppose* $n, d$ *are fixed but the number of prompts* $B \to \infty$. *Then* $\widetilde{V}_*$ *will converge to a limit* $\widetilde{V}_\infty$ *given by*

$$
\widetilde{V}_\infty = -\eta \left( \frac{n+1}{n} A + \frac{\mathrm{tr}(A)}{n} A A^\dagger \right)^\dagger - \Gamma_0^{-1}(I - A A^\dagger) \quad (4.2)
$$

## 5 Data Selection for Post-training via SFT

**Proposition 5.1.** *Consider an LSA model with parameters* $(\widetilde{V}, \widetilde{W})$. *We fix* $\widetilde{W} = I$ *and assume a test prompt of the form* $P = (x_1, \langle w, x_1 \rangle, \dots, x_m, \langle w, x_m \rangle)$. *Initializing the in-context learning with* $w_0 = 0$, *the predicted weight is given by* $\hat{w} = -\frac{1}{n} \widetilde{V} X X^\mathsf{T} w^*$ *with* $X = [x_1 | \dots | x_n] \in \mathbb{R}^{d \times n}$. *In addition, if* $x_i \sim \mathsf{N}(0, \Sigma)$, *we have*

$$
\mathbb{E}_{X, w^*}[\|\hat{w} - w^*\|^2] = \mathbb{E}_X\left[ \left\| I + \widetilde{V}\widehat{\Sigma} \right\|_F^2 \right] =
$$

$$
= \left\| I + \widetilde{V}\Sigma \right\|_F^2 + \frac{1}{n} \left( \mathrm{tr}(\widetilde{V}\Sigma^2 \widetilde{V}^\mathsf{T}) + \mathrm{tr}(\widetilde{V}\Sigma\widetilde{V}^\mathsf{T})\mathrm{tr}(\Sigma) \right)
$$
(5.1)

*where the expectation is with respect to randomness in* $X$ *and* $w^* \sim \mathsf{N}(0, I_d)$.

In the test error (5.1), we focus on the dominant term $\left\| I + \widetilde{V}\Sigma \right\|_F$ for large prompt length $n$. Assuming post-training features are i.i.d. from $\mathsf{N}(0, A)$ for some $A \succeq 0$, the post-training weights $\widetilde{V}_*(A)$ depend on the covariance $A$ via $\Phi$ in (4.1). Thus, optimal data selection reduces to choosing covariance $A$ that minimizes the post-test error.

### 5.1 Optimal Data Allocation

To analyze the interaction between pre-training and post-training, we consider the test-time covariance $\Sigma = \Sigma_0 +$

$\Delta$, where $\Sigma_0$ represents the distribution seen during pre-training and $\Delta$ denotes the adaptation task shift. We now characterize how the choice of the post-training covariance $A$ affects the post-test error across different subspaces.

Let $U = \text{range}(A)$. From (4.1), the term $\Phi$ shares the range $U$, while on the orthogonal complement $U^\perp$, the weight matrix $\tilde{V}_*$ acts simply as the pre-trained inverse $-\Gamma_0^{-1}$. Furthermore, outside the range of the adaptation shift $\Delta$, the test-time covariance $\Sigma$ coincides with the pre-training covariance $\Sigma_0$. Since $\Gamma_0^{-1}\Sigma_0 \approx I$ by the definition of $\Gamma_0$ in (3.7), the residual error $I + \tilde{V}\Sigma$ on $U^\perp$ becomes negligible if we align $U$ with $\text{range}(\Delta)$. This alignment ensures that the post-training resources are concentrated exclusively on the subspace where the pre-trained model exhibits a deficit.

Restricted to the adaptation subspace $\mathcal{U} = \text{range}(\Delta)$, the population-limit error operator is expressed as:

$$P_U(I + V_\infty\Sigma)P_U$$
$$= I - \eta\left(\frac{n+1}{n}A + \frac{\text{tr}(A)}{n}I\right)^{-1}(P_U\Sigma_0 P_U + \Delta)$$

In the high-dimensional regime (large $n$), the trace term and the $1/n$ scaling factors become secondary, implying that the optimal choice for the post-training covariance is approximately $A \approx \eta(P_U\Sigma_0 P_U + \Delta)$.

**Connection to example hardness.** In practice, post-training is often employed to address "gaps" in the model—specifically, skills or topics that were missing or underrepresented during pre-training. To capture such scenarios, we assume that the range of the pre-training covariance $\Sigma_0$ and the range of the adaptation shift $\Delta$ have a small inner product (i.e., they are nearly orthogonal). Consequently, $P_U\Sigma_0 P_U$ constitutes only a small component of $\Sigma_0$. We argue that in these scenarios, the most effective strategy is to select post-training examples that the pre-trained model finds "hard". Specifically, Proposition 5.1 establishes that the error of a pre-trained model on a task with prompts $x_{i,\tau} \sim \mathcal{N}(0, A)$ is approximately $\mathcal{L}_{pre} \approx \|I - \Gamma_0^{-1}A\|_F^2$. Because the support of $\Sigma_0$ is small on $\text{range}(\Delta)$, the operator $\Gamma_0^{-1}$—which essentially acts as the inverse of the pre-training density—takes its largest values on this space. Therefore, examples whose covariance is spanned by $\text{range}(\Delta)$ represent directions where the pre-trained model has the least confidence and highest residual error. This leads to our first key insight:

**Insight 1:** Selecting examples that are "hard" for the pre-trained model (i.e., those aligned with the adaptation shift $\Delta$) is the most effective strategy for post-training.

We next present an experiment designed to verify Insight 1. The pretraining distribution is $\mathsf{N}(0, \Sigma_0)$, where $\Sigma_0 = \text{diag}(\rho 1_m, 1_{d-m})$ with $d = 400$, $m = 200$, and $\rho = 0.2$. The post-test distribution is defined by $\Sigma = $

$\Sigma_0 + \Delta$, where $\Delta = \text{diag}(1_m, 0_{d-m})$. In this setting, the hardest test distributions for the pretrained model are those whose covariance lies in the range of $\Delta$. During post-training, data are sampled from $\mathsf{N}(0, A)$, where $A = \text{diag}(\cos(\theta)1_m, \sin(\theta)\sqrt{m/(d-m)}\,1_{d-m})$ and $\theta \in (0, \pi/2)$. Smaller values of $\theta$ correspond to stronger alignment with the shift $\Delta$, and therefore to harder test distributions for the pretrained model. The results are shown in Figure 1. In Figure 1a, we vary $B$ and report the test error in (5.1). The minimum test error over $B$ is attained at the smallest $\theta$, and increases as $\theta$ grows, which supports Insight 1 that harder examples are more beneficial for SFT. Figure 1b reports the test error as we vary $n$ (prompt length).

## 5.2 Data Scaling in SFT

We study how SFT data size affects post-training performance by analyzing the expected error (Proposition 5.1) on post-test prompts $\mathsf{N}(0, \Sigma)$. We examine how this error varies with the number of prompts $B$ and the prompt length $n$ during SFT.

We first present experiments, followed by theory supporting the resulting insights. The pretraining distribution is $\mathsf{N}(0, \Sigma_0)$ with $\Sigma_0 = \text{diag}(\rho 1_m, 1_{d-m})$, $d = 400$ and $m = 200$. The post-test distribution uses $\Sigma = \Sigma_0 + \Delta$, where $\Delta = \text{diag}(1_m, 0_{d-m})$. During post-training, data is drawn from $\mathsf{N}(0, A)$ with $A = \text{diag}(\eta(\rho + 1)1_m, r1_{d-m})$, matching $\eta\Sigma$ on the first $m$ coordinates and using $r$ on the rest. We set $\rho$ and $r$ small so the first $m$ directions are underrepresented in pretraining and can be strengthened during post-training. When $r = 0$, the post-train distribution matches the optimal allocation of Section 5.1. However, nonzero $r$ introduces interference between post-training and pretraining data, which is often the case in practice. By (4.1), the transformer parameters depend on the pseudo-inverse of the empirical covariance, so smaller nonzero $r$ yields stronger interference.

In the first experiment, we vary the number of prompts $B$ from 50 to 2000, for prompt lengths $n \in \{400, 800, 1200\}$, fix $\rho = 0.1$, and consider interference levels $r \in \{0, 0.01, 0.1\}$. Fig. 2 shows that the error exhibits double descent, with an overshoot at $B = m$ when $r = 0$ and at $B = d$ when $r \neq 0$. The error first decreases with $B$, then increases again, and the crossover point grows with the prompt length $n$. When interference is strong, the error remains above its value at optimal $B$ even in large $B$ limit (Fig. 2b). In the second experiment, we vary the prompt length $n$ from 20 to 1000 and evaluate post-test error at $B \in \{50, 150, 300, 500\}$. As shown in Figure 3, the error trends differ across choice of $B$. Under interference and for small to moderate values of $B$, it first decreases with $n$ and then becomes monotonically increasing, yielding a U-shaped curve and indicating an optimal prompt length that minimizes test error.

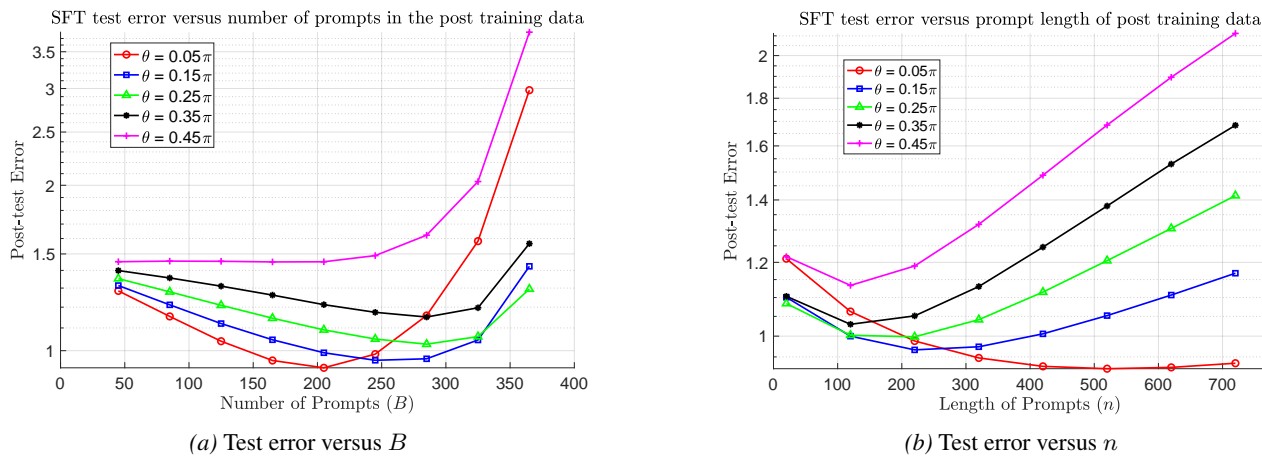

*Figure 1.* Post-test error as the number of prompts $B$, and the length of prompts $n$ vary, for different choices of $\theta$. Here, $d = 400, m = 200, \rho = 0.2$. Pre-trained covariance is $\Sigma_0 = \text{diag}(\rho 1_m, 1_{d-m})$, and the adaptation shift is $\Delta = \text{diag}(1_m, 0_{d-m})$. Here, $A = \text{diag}(\cos(\theta)1_m, \sin(\theta)\sqrt{m/(d-m)}1_{d-m})$, with $\theta \in (0, \pi/2)$, so smaller $\theta$ corresponds to harder distribution for the pretrained model.

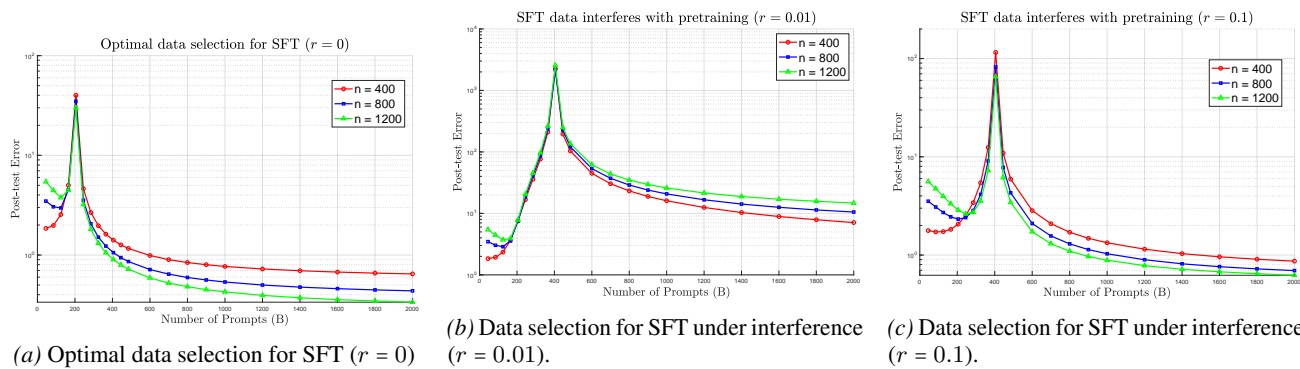

*Figure 2.* Post-test error as the number of prompts $B$ varies. Here, $d = 400, m = 200$ with different prompt lengths $(n)$. Pre-trained covariance is $\Sigma_0 = \text{diag}(\rho 1_m, 1_{d-m})$, $\Delta = \text{diag}(1_m, 0_{n-m})$. Left panel represents the optimal SFT data allocation with covariance $A = \text{diag}(\eta(\rho + 1)1_m, 0_{d-m})$, with $\rho = 0.1$. The right panel represents the case that SFT data distribution interferes with the pretraining distribution. Here, $A = \text{diag}(\eta(\rho + 1)1_m, r1_{n-m})$, with $r = 0.01$.

These results show that increasing SFT data volume—either the number of prompts $B$ or the prompt length $n$—can paradoxically degrade performance in the presence of interference. The key trade-off is that more SFT data helps the model learn underrepresented dimensions from pretraining, but also amplifies interference that erodes pretrained capabilities. Our findings therefore suggest an optimal data size that balances these competing effects. This further supports the empirical preference for small, high-quality datasets, whose high information density enables effective adaptation without the catastrophic costs of over-parameterization and interference. We formalize this observation as follows:

**Insight 2:** To mitigate the effects of interference between pretraining and post-training, SFT datasets should be curated to be relatively small in volume and high in quality.

In Appendix B we analyze the post-test error. The analysis, consistent with our experiments, predicts that the test error

diverges as $B \to d$ when interference is present $(r \neq 0)$ and as $B \to m$ when $r = 0$. We further characterize the asymptotic limit of the post-test error in the scaling regime where $d, m$, and $B \to \infty$ while their relative ratios remain constant. This analysis demystifies the quantitative effect of different factors on the test error behavior.

*Remark* 5.2 (Effect of $r$). When $r = 0$, the pretrained subspace lies in the null space of the feature matrix, and the pseudo-inverse removes it, so there is no interference. For small $r > 0$, weak overlap between the pretrained features and the training data leads to near-zero eigenvalues in the covariance. Since the interpolating estimator must invert these small eigenvalues to fit the training data, the learned weights become larger, which increases variance and, in turn, test error. In contrast, larger values of $r$ improve conditioning, allowing the pretrained features to be incorporated more stably and effectively. This behavior is consistent with the theory in Appendix B, where the test error contains terms

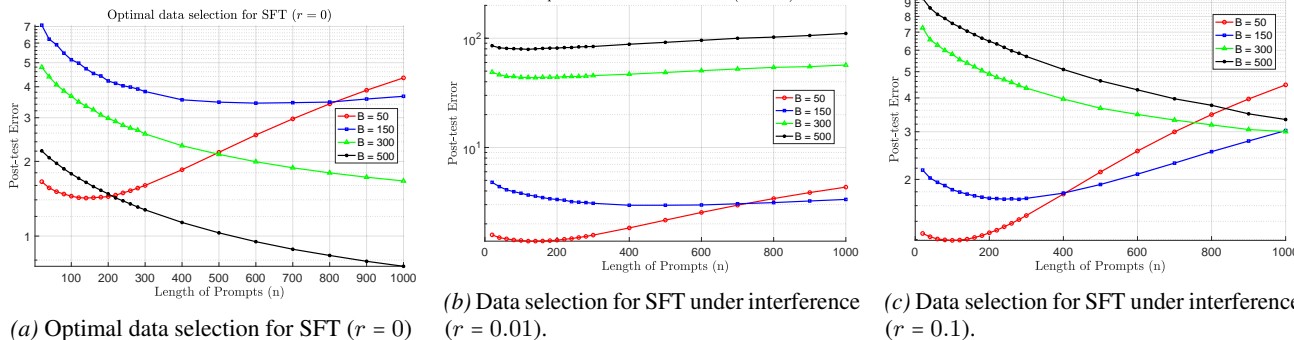

*(a)* Optimal data selection for SFT ($r = 0$)

*(b)* Data selection for SFT under interference ($r = 0.01$).

*(c)* Data selection for SFT under interference ($r = 0.1$).

*Figure 3.* Behavior of the post-test error as we varying the prompt length $n$, under the same setup as in Figure 2.

proportional to $1/r$.

It is worth noting that in the experiment reported in Figure 1, the minimum test error for each $\theta$ is attained at intermediate values of $B$ and $n$, which is consistent with Insight 2 that SFT data should be of moderate size.

## 6    Analysis of the OS loss

We begin by deriving a more direct characterization of the outcome supervision (OS) loss.

**Proposition 6.1.** *For the LSA model with k-step of thinking during the post-training the OS loss can be written as*

$$\mathcal{L}^{\mathrm{OS}}(\widetilde{V}, \widetilde{W}) = \frac{1}{2B}\sum_{\tau=1}^{B}\left\|\left(I + \sum_{i=0}^{k-1}(\widetilde{V}S_\tau\widetilde{W} + I)^i\widetilde{V}S_\tau\right)w_\tau^*\right\|_{\ell_2}^2$$

The parameters $(\widetilde{V}, \widetilde{W})$ are initialized at $(-\Gamma_0^{-1}, I)$ from the pretraining stage. We next study the landscape of the OS loss which demystifies several intriguing characteristics of post-training via OS and how it compares with SFT post training. To simplify our discussions and derivations, we fix $\widetilde{W} = I$ and only update $\widetilde{V}$ via gradient descent. However, we expect our discussion to extend to the general case of updating both parameters, albeit with a more complicated derivations. In our experiments, we update all of the transformer weights and showing our insights from analysis are empirically observed as well.

By fixing $\widetilde{W} = I$, the OS loss simplifies to:

$$\mathcal{L}^{\mathrm{OS}}(\widetilde{V}, I) = \frac{1}{2B}\sum_{\tau=1}^{B}\left\|(I + \widetilde{V}S_\tau)^k w_\tau^*\right\|_{\ell_2}^2.$$

Let $M_\tau = I + \widetilde{V}S_\tau$. As derived in Appendix E, the gradient of the OS loss with respect to the operator $V$ is given by:

$$\nabla_V\mathcal{L}^{\mathrm{OS}} = \frac{1}{B}\sum_{\tau=1}^{B}\sum_{j=0}^{k-1}(M_\tau^T)^j M_\tau^k w_\tau^*(w_\tau^*)^T(M_\tau^T)^{k-1-j}S_\tau^T.$$

**Vanishing and growing gradients in OS Loss.** The gradient contains the term $M_\tau^k$, which acts as a powerful scaling factor. In the stable region ($\rho(M_\tau) < 1$), the term $M_\tau^k$ shrinks the gradient toward zero exponentially fast as the chain length $k$ increases. In this regime, the model is already stable on the task, but the vanishing gradient makes it increasingly difficult to "nudge" the matrix $\widetilde{V}$ into the optimal subspace for further refinement. Conversely, if $\rho(M_\tau) > 1$, the gradient has an exponential growth in $k$. This creates a sharp "cliff" in the loss landscape near the edge of stability ($\rho \approx 1$), and training requires infinitesimally small step sizes to prevent numerical divergence.

**Sharpness and curvature of the landscape.** Because the OS loss is effectively a degree-$2k$ polynomial, the Hessian $\nabla^2\mathcal{L}$ is highly sensitive to the operator's spectral properties. As shown in Appendix E, near a global minimum where $M_\tau^k w_\tau^* \approx 0$, the Hessian spectral norm $\lambda_{\max}$ scales as:

$$\lambda_{\max}(H) \propto \frac{1}{B}\sum_{\tau=1}^{B}k^2 \cdot \rho(M_\tau)^{2k-2} \tag{6.1}$$

This indicates that the curvature grows quadratically with the number of iterations $k$ near the boundary of stability. If gradient descent is not run for a sufficient duration, the model remains near this high-curvature "cliff." In this state, small variations—arising from finite $n$, $B$, or sample noise during post-test evaluations—can push the model back into the unstable region, leading to "overthinking", even if it pulled into the stable region during training.

**Insight 3:** The sharp curvature near $\rho \approx 1$ suggests that Outcome Supervision (OS) is prone to instability unless trained with large amounts of data $(n, B)$ and many gradient steps. Insufficient training leaves the model at a "sharp" minimum where minor distribution shifts cause large errors. Nonetheless, with access to a sufficiently large data pool, OS can acquire new skills beyond those already encoded in the pretrained model.

**Pretraining and Generalization.** The pretrained model, which serves as the initialization for the OS loss, plays a

critical role in OS stability. Consider a new task drawn from the test-time covariance $\Sigma = \Sigma_0 + \Delta$, with $\Sigma_0$ the pretraining covariance and $\Delta$ the adaptation shift. Near initialization, and assuming a sufficiently large prompt length $n$ such that $S_\tau \to \Sigma$, the learned operator $V$ is dominated by the prior $V_0 \approx -\Gamma_0^{-1}$. Consequently, we have $VS_\tau \approx -\Gamma_0^{-1}(\Sigma_0 + \Delta) \approx -I - \Gamma_0^{-1}\Delta$. Thus, the transition matrix becomes:

$$M_\tau = I + VS_\tau \approx -\Gamma_0^{-1}\Delta \implies \rho(M_\tau) \approx \rho(\Gamma_0^{-1}\Delta).$$

This relationship reveals two distinct optimization regimes based on the spectral alignment between the pretraining distribution and the adaptation shift:

- **Case 1: Incremental adaptation (spectral alignment).** When $\Gamma_0$ is large in the directions where $\Delta$ is prominent—implying the pretraining distribution effectively covers the shift—the spectral radius $\rho(M_\tau)$ remains small. In this regime, the model initializes within the stable region ($\rho < 1$), permitting a safe, albeit gradual, refinement of the model parameters.

- **Case 2: New task adaptation (spectral misalignment).** If the task involves novel subspaces where $\Gamma_0$ is small but $\Delta$ is large, the spectral radius becomes large, i.e., $\rho(\Gamma_0^{-1}\Delta) \gg 1$. The model starts deep in the unstable region, requiring a drastically reduced step size $\eta$ to maintain stability:

$$\eta < \frac{2}{\lambda_{\max}(H)} \propto \frac{C}{k^2 \rho(M_\tau)^{2k-2}},$$

by (6.1). These observations are summarized below:

**Insight 4: Synergy of pretraining and Outcome Supervision.** OS is most effective at improving performance on tasks already partially learned during pretraining. For novel tasks, the high initial spectral radius necessitates a slow and potentially unstable training procedure.

**Practical Implications for Training.** Given the strong similarities between OS and RL, we expect Insight 4 to extend naturally to RL settings. Stability imposes additional constraints in both outcome supervision and RL. To ensure the eigenvalues remain within the stable regime, the learning rate must be carefully tuned to the sharpest direction of the Hessian. This creates a stark disparity in the optimization landscape: the step size $\eta$, forced to be infinitesimally small by the unstable directions, can be too small to make meaningful progress in the data-aligned directions. In addition, while RL does not require the high-quality, human-curated labels necessary for SFT, it compensates by requiring massive data diversity and volume. A large number of gradient steps is needed to overcome the slow progress in "flat" directions, while a high volume of data ensures the model is pushed deep into the stable region across a broad spectrum of tasks, reducing the risk of "overthinking" during inference.

## 7 Data Diversity and Distributional Balance in Pretraining

In our analysis, the influence of the pretrained model on post-trained model is mathematically encapsulated in the initialization $V_0 = -\Gamma_0^{-1}$, where by definition (3.7), $\Gamma_0 \approx \Sigma_0$ the pretraining covariance. The post-test error, characterized by Proposition 5.1, is governed by the product $V\Sigma = V(\Sigma_0 + \Delta)$; at initialization, this yields $V\Sigma \approx -I - \Gamma_0^{-1}\Delta$. Consequently, an imbalanced pretraining distribution—characterized by a singular or ill-conditioned $\Gamma_0$—imposes a severe penalty on adaptation in new directions where $\Gamma_0$ is small but $\Delta$ is large. While SFT can partially mitigate a misaligned prior through the stabilizing influence of supervised signals, the OS and RL optimization is strictly bottlenecked by the spectral alignment between $\Gamma_0$ and $\Delta$. If $\Gamma_0$ lacks sufficient diversity, even minor shifts in novel subspaces trigger an exponential escalation of the Hessian's spectral norm, scaling as $k^2\rho^{2k-2}$. This spectral divergence necessitates infinitesimally small step sizes and renders the model sensitive to variations in sample prompts in training. Such instability often manifests as "overthinking" during inference. Therefore, pretraining must prioritize distributional balance and data diversity as essential mechanisms for optimization stability. A broad spectral prior ensures the model initializes within the stable regime ($\rho < 1$), effectively smoothing the high-curvature "cliffs" of the RL landscape into manageable, flat regions for downstream adaptation.

## 8 Experiments

In this section, we conduct experiments to validate our theoretical results. We refer to Appendix F.2 for additional experiments on Qwen 2.5-7B-Instruct on the M23k dataset (Huang et al., 2025b), which illustrate the practical relevance of our analysis.

**Setting.** We conduct experiments in two settings. First, we consider a transformer with a single linear self-attention (LSA) to confirm the results of our theorems. Then, we consider large, nonlinear transformer architecture namely GPT2 to validate the generality of our conclusions. Due to space constraints, we defer discussion of the LSA experiments to Appendix F.1. These experiments exhibit trends similar to the GPT-2 results presented here, and are consistent with our theoretical insights. In both sets of experiments, the data distribution follows our in-context weight prediction task in Sec. 3, where in the pre-training, data has a covariance of $\Sigma_0$, and in the post-testing with SFT or OS we have $\Sigma = \Sigma_0 + \Delta$. During post-training, we let the model to output multiple steps before returning the final predicted weight vector, i.e., at each step $i$ we concatenate the embedding with $[0_d, \hat{w}_i, 1]$ as in Eq. (3.3) and input the concatenated embedding matrix to the model. The estimated

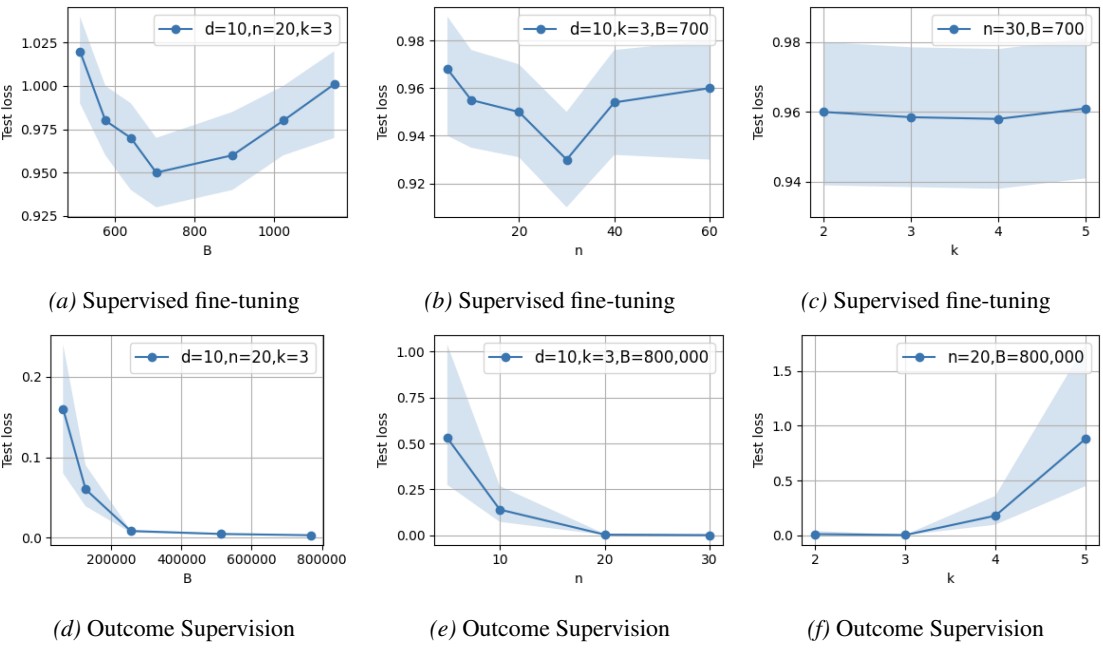

*Figure 4.* GPT-2 experiments: Test loss for (a)-(c) post-training with SFT, and (d)-(f) post-training with Outcome Supervision (OS). For SFT, there is a turning point where larger sample size ($B$) and context-length ($n$) hurt the performance. In contrast, for OS larger $B, n$ improves the performance.

$\hat{w}_k$ will be returned after $k$ steps of Chain of Thought (CoT). We report the average results and error bars over 10 runs.

**Pretrain, post-train, and test data.** We generate pretraining data using $\Sigma_0$ where $\Sigma_{i,i} = 0.1$ for $i \in \{1, \ldots, d/5\}$ and $\Sigma_{i,i} = 1$ for $i \in \{d/5, \ldots, d\}$. Then, we post-train the transformer on the synthetic data generated with $\Delta$, where $\Delta$ is a low rank PSD matrix with $\Delta_{i,i} = 1$. For testing the model, we use $\Sigma = \Sigma_0 + \Delta$.

**Large, nonlinear transformer architectures.** We use a decoder-only Transformer architecture (Vaswani et al., 2017) from the GPT-2 family (Radford et al., 2019), consisting of 12 layers, 8 attention heads and a 256-dimensional embedding space. In total model contains 9.5M parameters. This architecture takes as input a sequence of vectors in its embedding space and predicts the weight vector within the same space. We apply this architecture to prompts of form $(x_{\tau,1}, y_{\tau,1}, \cdots, x_{\tau,m}, y_{\tau,m}, w_0, 1)$ in the following manner. In line with (Garg et al., 2022), we map each $y_{\tau,i}$ to the same dimension as $x_{\tau,i}$ by appending zeros, and map $x_{\tau,i}, y_{\tau,i}$ into the latent embedding space of the Transformer through a (learnable) linear transformation. We get the predicted $w_\tau$ as the model output. Similarly, we map the model output, i.e., $w_\tau$ from the latent embedding space of the Transformer to a d-dimensional vector through another (learnable) linear transformation. Training is performed with a batch size of 64 over 100 steps for SFT and $12k$ steps for OS. The model is first pretrained with a CoT length $k = 8$. During both training and test, we apply CoT with length $k = 3$. We used

curriculum learning (Garg et al., 2022) to speed up training.

Fig. 4 (a)-(c) show the results when post-training is done with the SFT loss. Fig. 4a,4b show that increasing the sample size ($B$) or context length ($n$) initially yields a lower test loss but further increasing the sample size or context length increases the test loss. Fig. 4c shows that the test loss is relatively robust and not sensitive to the length of post-training CoT ($k$). Fig 4 (d)-(f) show the results when post-training is done with the OS loss. In contrast to SFT, we see that OS benefits from larger sample size ($B$) and context length ($n$). In addition, longer CoT ($k$) during post-training increases the test loss and degrades the performance, confirming insight 4 in Section 6.

## 9  Conclusion

Our work provides a theoretical and empirical framework for jointly designing pretraining and post-training for LLMs. Balanced pretraining creates latent capabilities best activated by SFT on small numbers of carefully selected, hard examples aligned with the target shift. Scaling up SFT data introduces interference that erodes pretrained structure, favoring small, high-quality datasets. Outcome Supervision and RL have a sharply curved, unstable landscape that make them data-hungry, yet effective for refining partially learned pretrained capabilities. These insights guide optimal combined use: targeted SFT for efficient adaptation on challenging examples, complemented by large-scale RL (Outcome Supervision) for robust skill refinement.

## Impact Statement

This paper presents work whose goal is to advance the field of Machine Learning. There are many potential societal consequences of our work, none of which we feel must be specifically highlighted here.

## Acknowledgments

AJ was supported in part by the NSF Award DMS-2311024, an Amazon Faculty Research Award, an Adobe Faculty Research Award, and an iORB grant form USC Marshall School of Business. BM was supported in part by the NSF CAREER Award 2146492, NSF-Simons AI Institute for Cosmic Origins (CosmicAI) and NSF AI Institute for Foundations of Machine Learning (IFML).

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

# A    Proof of theorems and technical lemmas

## A.1    Proof of Theorem 4.1

As $\lambda \to 0^+$, the minimizer $(\widetilde{V}_\lambda, \widetilde{W}_\lambda)$ must converge to a point $(\widetilde{V}_*, \widetilde{W}_*)$ in the zero-loss manifold of $L(\widetilde{V}, \widetilde{W})$ that is closest to the initialization $(-\Gamma_0^{-1}, I)$ in the Frobenius norm.

We first simplify the dynamic of LSA into a recurrent update on the estimated weight $\hat{w}_i$. We have We have

$$
f_{\mathrm{LSA}}(Z_i, \theta^*)_{[:, -1]} = \begin{bmatrix} 0_{d\times 1} \\ 0 \\ \hat{w}_i \\ 1 \end{bmatrix} + V Z_i \cdot \frac{Z_i^\top W Z_{i[:, -1]}}{n}
$$

$$
= \begin{bmatrix} 0_{d\times 1} \\ 0 \\ \hat{w}_i \\ 1 \end{bmatrix} + \frac{1}{n} V Z_i Z_i^\top \begin{bmatrix} \widetilde{W}\hat{w}_i \\ -1 \\ 0 \\ 0 \end{bmatrix}
$$

$$
= \begin{bmatrix} 0_{d\times 1} \\ 0 \\ \hat{w}_i \\ 1 \end{bmatrix} + \frac{1}{n} \begin{bmatrix} 0_{d\times n} & 0_{d\times 1} & 0_{d\times 1} & 0_{d\times 1} \\ 0_{1\times n} & 0 & 0 & 0 \\ \widetilde{V}X & 0_{d\times 1} & 0_{d\times 1} & 0_{d\times 1} \\ 0_{1\times n} & 0 & 0 & 0 \end{bmatrix} \begin{bmatrix} X & 0 & 0 & \dots & 0 \\ y & 0 & 0 & \dots & 0 \\ 0_{d\times n} & w_0 & \hat{w}_1 & \dots & \hat{w}_i \\ 0_{1\times n} & 1 & 1 & \dots & 1 \end{bmatrix}^\top \begin{bmatrix} \widetilde{W}w_0 \\ -1 \\ 0 \\ 0 \end{bmatrix}
$$

$$
= \begin{bmatrix} 0_{d\times 1} \\ 0 \\ \hat{w}_i \\ 1 \end{bmatrix} + \frac{1}{n} \begin{bmatrix} 0_{d\times n} & 0_{d\times 1} & 0_{d\times 1} & 0_{d\times 1} \\ 0_{1\times n} & 0 & 0 & 0 \\ \widetilde{V}X & 0_{d\times 1} & 0_{d\times 1} & 0_{d\times 1} \\ 0_{1\times n} & 0 & 0 & 0 \end{bmatrix} \begin{bmatrix} X^\top \widetilde{W}\hat{w}_i - y^\top \\ 0 \end{bmatrix}
$$

$$
= \begin{bmatrix} 0_{d\times 1} \\ 0 \\ \hat{w}_i \\ 1 \end{bmatrix} + \frac{1}{n} \begin{bmatrix} 0_{d\times 1} \\ 0 \\ \widetilde{V}X X^\top (\widetilde{W}\hat{w}_i - w^*) \\ 0 \end{bmatrix}.
$$

Hence, we obtain the following recursions for each of the prompt weight vectors:

$$
\hat{w}_{i+1, \tau} = \hat{w}_{i, \tau} + \widetilde{V} S_\tau (\widetilde{W}\hat{w}_{i, \tau} - w^*_\tau). \tag{A.1}
$$

Now note that in the SFT loss, at each step we give the model the CoT ground-truth sequence $(w_{1, \tau}, \dots, w_{i, \tau})$ and compute the error $\|w_{i+1, \tau} - \hat{w}_{i, \tau}\|_{\ell_2}^2$. Let $\rho = 1 - \eta$. Given $w_{i, \tau} = (1 - \rho^i) w^*_\tau$, we define the residual $R_{i, \tau}$ for $i = 0, \dots, k$ and $\tau = 1, \dots, B$ as follows:

$$
\begin{aligned}
R_{i, \tau} &= w_{i, \tau} + \widetilde{V} S_\tau (\widetilde{W} w_{i, \tau} - w^*_\tau) - w_{i+1, \tau} \\
&= (1 - \rho^i) w^*_\tau + \widetilde{V} S_\tau (\widetilde{W}(1 - \rho^i) w^*_\tau - w^*_\tau) - (1 - \rho^{i+1}) w^*_\tau \\
&= \widetilde{V} S_\tau (\widetilde{W} - I) w^*_\tau - \rho^i (\widetilde{V} S_\tau \widetilde{W} + \eta I) w^*_\tau
\end{aligned}
$$

We characterize this manifold by analyzing the residual $R_{i, \tau}$ for each block $\tau$ and iteration $i \in \{0, \dots, k\}$. The loss function can be written as

$$
\mathcal{L}^{\mathrm{SFT}}(\widetilde{V}, \widetilde{W}) = \frac{1}{2B} \sum_{\tau=1}^{B} \sum_{i=0}^{k} \|R_{i, \tau}\|_{\ell_2}^2.
$$

To characterize the zero-loss manifold, note that for $L(\widetilde{V}, \widetilde{W}) = 0$, we require $R_{i, \tau} = 0$ for all $i$. Since $1$ and $\rho^i$ are linearly independent for $i \neq 0$, the coefficients of the polynomial in $\rho^i$ must vanish independently:

1. $\widetilde{V} S_\tau (\widetilde{W} - I) w^*_\tau = 0$

2. $(\widetilde{V} S_\tau \widetilde{W} + \eta I) w^*_\tau = 0 \implies \widetilde{V} S_\tau \widetilde{W} w^*_\tau = -\eta w^*_\tau$

Substituting the second condition into the first, we obtain:

$$\widetilde{V}S_\tau\widetilde{W}w_\tau^* - \widetilde{V}S_\tau w_\tau^* = 0 \implies -\eta w_\tau^* - \widetilde{V}S_\tau w_\tau^* = 0 \implies \widetilde{V}S_\tau w_\tau^* = -\eta w_\tau^*,$$

for all $\tau = 1, \ldots, B$. Let $\Omega = [w_1^*, \ldots, w_B^*]$ and $\Phi = [S_1 w_1^*, \ldots, S_B w_B^*]$. The system is expressed as $\widetilde{V}\Phi = -\eta\Omega$. The limit $\widetilde{V}_*$ minimizes $\left\|\widetilde{V} + \Gamma_0^{-1}\right\|_F^2$ subject to $\widetilde{V}\Phi = -\eta\Omega$, which is solved via the Moore-Penrose pseudoinverse:

$$\widetilde{V}_* = -\eta\Omega\Phi^\dagger - \Gamma_0^{-1}(I - \Phi\Phi^\dagger).$$

The term $(I - \Phi\Phi^\dagger)$ is the orthogonal projection onto the null space of $\Phi^\top$, ensuring $\widetilde{V}$ follows the initialization $-\Gamma_0^{-1}$ in directions not spanned by the data.

Now that $\widetilde{V}_*$ is characterized, we proceed with proving that $\widetilde{W}_* = I$. Note that this choice of $\widetilde{V}_*, \widetilde{W}_*$ satisfies both of the gradient condition (1) and (2) above. In addition, due to the penalty $\lambda\left\|\widetilde{W} - I\right\|_F^2$, we get $\widetilde{W}_* = I$ as the unique minimizer.

## A.2 Proof of Theorem 4.2

Let $\rho = 1 - \eta$ and $c_k = \sum_{i=0}^k \rho^{2i}$. Given $\widetilde{W} = I$ and $w_{i,\tau} = (1 - \rho^i)w_\tau^*$, the residual is $R_{i,\tau} = -\rho^i(\widetilde{V}S_\tau + \eta I)w_\tau^*$ and the loss can be written as

$$\mathcal{L}^{\mathrm{SFT}}(\widetilde{V}, I) = \frac{c_k}{2B}\left\|\widetilde{V}\Phi + \eta\Omega\right\|_F^2,$$

where we recall $\Phi = [S_1 w_1^*, \ldots, S_B w_B^*]$ and $\Omega = [w_1^*, \ldots, w_B^*]$. The gradient of the loss is given by

$$\nabla_{\widetilde{V}}\mathcal{L}_{SF} = \frac{c_k}{B}(\widetilde{V}\Phi + \eta\Omega)\Phi^\top$$

Defining $\Delta_t = \widetilde{V}_t - \widetilde{V}_*$ and noting $\widetilde{V}_*\Phi = -\eta\Omega$, the GD update $\widetilde{V}_{t+1} = \widetilde{V}_t - \gamma\nabla_{\widetilde{V}}\mathcal{L}_{SF}(\widetilde{V}_t, I)$ yields:

$$\Delta_{t+1} = \Delta_t\left(I - \frac{\gamma c_k}{B}M\right), \quad M = \Phi\Phi^\top$$

The error norm evolves as $\|\Delta_{t+1}\|_F \le \|\Delta_t\|_F \cdot \left\|I - \frac{\gamma c_k}{B}M\right\|_{\mathrm{op}}$, with $\|\cdot\|_{\mathrm{op}}$ indicating the operator norm.

Note that the condition $\gamma < \frac{2B}{c_k\lambda_{\max}(M)}$ ensures that $\left\|I - \frac{\gamma c_k}{B}M\right\|_{\mathrm{op}} < 1$ and so the GD updates converges to $\widetilde{V}_*$. Specifically, the contraction factor is determined by the most extreme eigenvalues that the error $\Delta_t$ sees in the subspace spanned by the data $\Phi$. On the range of $\Phi$, the contraction factor is given by

$$\alpha := \max\left(\left|1 - \frac{\gamma c_k}{B}\lambda_{\max}(M)\right|, \left|1 - \frac{\gamma c_k}{B}\lambda_{\min}^+(M)\right|\right)$$

By choosing $\gamma = \frac{B}{c_k\lambda_{\max}(M)}$, the rate simplifies to

$$\alpha = 1 - \frac{\lambda_{\min}^+(M)}{\lambda_{\max}(M)}$$

Substituting $\Delta_0 = \widetilde{V}_0 - \widetilde{V}_* = -\Gamma_0^{-1} - \widetilde{V}_*$, we obtain the desired bound:

$$\|\widetilde{V}_t - \widetilde{V}_*\|_F \le \left(1 - \frac{\lambda_{\min}^+(M)}{\lambda_{\max}(M)}\right)^t \|\Gamma_0^{-1} + \widetilde{V}_*\|_F,$$

which completes the proof.

## A.3 Proof of Proposition 4.4

Recalling from (4.1), $V_*$ satisfies the system $\widetilde{V}\Phi = -\eta\Omega$. To find the explicit limit as $B \to \infty$, we analyze the normal equations:

$$\widetilde{V}\left(\frac{1}{B}\Phi\Phi^\top\right) = -\frac{\eta}{B}\Omega\Phi^\top$$

Recall $w_\tau^* \sim \mathcal{N}(0, I)$ and $S_\tau$ being the empirical covariance of $n$ samples from $\mathcal{N}(0, A)$. In addition, $w_\tau^*$ and $S_\tau$ are independent.

We have

$$\mathbb{E}\left[\frac{1}{B}\Omega\Phi^\top\right] = \mathbb{E}\left[\frac{1}{B}\sum_{\tau=1}^B w_\tau^*(S_\tau w_\tau^*)^\top\right] = \mathbb{E}[w^* w^{*\top} S_\tau^\top]$$

By independence and the fact that $\mathbb{E}[w^* w^{*\top}] = I$ and $\mathbb{E}[S_\tau] = A$, we get

$$\mathbb{E}\left[\frac{1}{B}\Omega\Phi^\top\right] = A$$

In addition,

$$\mathbb{E}\left[\frac{1}{B}\Phi\Phi^\top\right] = \mathbb{E}\left[\frac{1}{B}\sum_{\tau=1}^B (S_\tau w_\tau^*)(S_\tau w_\tau^*)^\top\right] = \mathbb{E}[S_\tau w^* w^{*\top} S_\tau^\top] = \mathbb{E}[S_\tau^2]$$

Using the properties of the Wishart distribution for $S_\tau = \frac{1}{n}\sum_{i=1}^n x_i x_i^\top$ with $x_i \sim \mathcal{N}(0, A)$, (see Lemma A.2 in (Javanmard et al., 2025)) we have

$$\mathbb{E}[S_\tau^2] = \frac{n+1}{n}A^2 + \frac{1}{n}\mathrm{tr}(A)A$$

First consider the case $A$ is invertible. By Slutsky's Theorem and the consistency of the sample covariance, as $B \to \infty$, the learned operator $\widetilde{V}$ converges in probability to:

$$\widetilde{V}_\infty = -\eta A \left(\mathbb{E}[S_\tau^2]\right)^{-1}$$

Substituting the explicit form of $\mathbb{E}[S_\tau^2]$:

$$\widetilde{V}_\infty = -\eta A \left(\frac{n+1}{n}A^2 + \frac{\mathrm{tr}(A)}{n}A\right)^{-1} = -\eta \left(\frac{n+1}{n}A + \frac{\mathrm{tr}(A)}{n}I\right)^{-1}$$

When $A$ is singular, the same derivation holds in the range of $A$. In the null space of $A$, $\widetilde{V}_\infty$ stays at its initialization $-\Gamma_0^{-1}$. Both cases can be unified as follows:

$$\widetilde{V}_\infty = -\eta \left(\frac{n+1}{n}A + \frac{\mathrm{tr}(A)}{n}AA^\dagger\right)^\dagger - \Gamma_0^{-1}(I - AA^\dagger),$$

which completes the proof.

### A.4 Proof of Proposition 5.1

Specializing the recursion (A.1) to $i = 0$ and $\widehat{W} = I$, we have $\hat{w} = w_0 + \frac{1}{n}\widetilde{V}XX^\top(w_0 - w^*)$. By choosing the initialization $w_0 = 0$ we arrive at $\hat{w} = -\frac{1}{n}\widetilde{V}XX^\top w^*$.

Letting $\widehat{\Sigma} = \frac{1}{n}XX^\top$, we have

$$\mathbb{E}[\|\hat{w} - w^*\|_{\ell_2}^2] = \mathbb{E}[\|I + \widetilde{V}\widehat{\Sigma}\|_F^2] = \|I + \widetilde{V}\Sigma\|_F^2 + \frac{1}{n}\left(\mathrm{tr}(\widetilde{V}\Sigma^2\widetilde{V}^\top) + \mathrm{tr}(\widetilde{V}\Sigma\widetilde{V}^\top)\mathrm{tr}(\Sigma)\right)$$

where the last step follows from Lemma A.1 below.

**Lemma A.1.** *Let $X = [x_1|\ldots|x_n]^\top$ with $x_i \sim \mathsf{N}(0, \Sigma)$ with $\Sigma \in \mathbb{R}^{d \times d}$. Define $\widehat{\Sigma} := \frac{1}{n}X^\top X$. Then, for any matrix $A \in \mathbb{R}^{d \times d}$, we have*

$$\mathbb{E}[\|I + A\widehat{\Sigma}\|_F^2] = \|I + A\Sigma\|_F^2 + \frac{1}{n}\left(\mathrm{tr}(A\Sigma^2 A^\top) + \mathrm{tr}(A\Sigma A^\top)\mathrm{tr}(\Sigma)\right) \tag{A.2}$$

*Proof.* (Proof of Lemma A.1) We write

$$\mathbb{E}[\lVert I + A\widehat{\Sigma} \rVert_F^2] = d + \mathbb{E}[\lVert A\widehat{\Sigma} \rVert_F^2] - 2\,\mathbb{E}[\mathrm{tr}(A\Sigma)] \tag{A.3}$$

From (Javanmard et al., 2025)(Lemma A.2) we have

$$\mathbb{E}[\widehat{\Sigma}(A^\mathsf{T}A)\widehat{\Sigma})] = \frac{n-1}{n}\Sigma(A^\mathsf{T}A)\Sigma + \frac{1}{n}\left(2\Sigma(A^\mathsf{T}A)\Sigma + \mathrm{tr}(\Sigma A^\mathsf{T}A)\Sigma\right).$$

Hence, by taking the trace of both sides and changing the orde of expectation and trace (since it is a linear operator), we get

$$\mathbb{E}[\lVert A\widehat{\Sigma} \rVert_F^2] = \frac{n+1}{n}\mathrm{tr}(A\Sigma^2 A^\mathsf{T}) + \frac{1}{n}\mathrm{tr}(A\Sigma A^\mathsf{T})\mathrm{tr}(\Sigma).$$

Here we also used the identity $\mathrm{tr}(AB) = \mathrm{tr}(BA)$ for square matrices of the same size.

Substituting back in (A.3) we obtain

$$\mathbb{E}[\lVert I + A\widehat{\Sigma} \rVert_F^2] = d + \mathrm{tr}(A^\mathsf{T}\Sigma^2 A) - 2\,\mathbb{E}[\mathrm{tr}(A\Sigma)] + \frac{1}{n}\left(\mathrm{tr}(A\Sigma^2 A^\mathsf{T}) + \mathrm{tr}(A\Sigma A^\mathsf{T})\mathrm{tr}(\Sigma)\right)$$

$$= \lVert I + A\Sigma \rVert_F^2 + \frac{1}{n}\left(\mathrm{tr}(A\Sigma^2 A^\mathsf{T}) + 2\mathrm{tr}(A\Sigma A^\mathsf{T})\mathrm{tr}(\Sigma)\right)$$

which completes the proof of lemma. ∎

## A.5   Proof of Proposition 6.1

We begin by recalling the recursion (A.1):

$$\hat{w}_{i+1,\tau} = \hat{w}_{i,\tau} + \widetilde{V}S_\tau(\widetilde{W}\hat{w}_{i,\tau} - w_\tau^*)$$
$$= (I + \widetilde{V}S_\tau\widetilde{W})\hat{w}_{i,\tau} - \widetilde{V}S_\tau w_\tau^*$$

Solving this recursion, we obtain

$$\hat{w}_{k,\tau} = (I + \widetilde{V}S_\tau\widetilde{W})^k \hat{w}_0 - \sum_{i=0}^{k-1}(I + \widetilde{V}S_\tau\widetilde{W})^i \widetilde{V}S_\tau w_\tau^*. \tag{A.4}$$

Next, using that $\hat{w}_0 = w_0 = 0$, we get

$$\mathcal{L}^{\mathrm{OS}}(V,W) = \frac{1}{2B}\sum_{\tau=1}^{B} \lVert \hat{w}_{\tau,k} - w_\tau^* \rVert_{\ell_2}^2$$
$$= \frac{1}{2B}\sum_{\tau=1}^{B} \left\lVert \left(I + \sum_{i=0}^{k-1}(\widetilde{V}S_\tau\widetilde{W} + I)^i \widetilde{V}S_\tau\right)w_\tau^* \right\rVert_{\ell_2}^2,$$

which completes the proof.

## B   Asymptotic Analysis of SFT post-training

We recall our notations from Section 4. Let $S_\tau := \frac{1}{n}\sum_{i=1}^{n} x_{i,\tau} x_{i,\tau}^\mathsf{T}$ be the empirical features covariance for $\tau = 1, \ldots, B$. We also define the following matrices:

$$\Omega := [w_1^*, \ldots, w_B^*] \in \mathbb{R}^{d\times B}, \quad \Phi := [S_1 w_1^*, \ldots, S_B w_B^*] \in \mathbb{R}^{d\times B}, \tag{B.1}$$

Also recall that the SFT data are generated as $x_i \sim \mathsf{N}(0, A)$ where $A = \eta(P_U\Sigma_0 P_U + \Delta) + rP_{U^\perp}$, with $U = \mathrm{range}(\Delta)$. When $r = 0$ this corresponds to the optimal data allocation discussed in Section 5.1 and $r \neq 0$ models the interference between SFT data and the pretrained model.

We consider the following specific structure for the pretrained covariance $\Sigma_0$ and distribution shift covariance $\Delta$ similar to our experiments in Section 5.2, namely

$$\Sigma_0 = \mathrm{diag}(\rho 1_m, 1_{d-m}), \qquad \Delta = \mathrm{diag}(1_m, 0_{d-m}).$$

During post-training, SFT data is generated from $\mathsf{N}(0, A)$ with

$$A = \mathrm{diag}(\eta(\rho + 1)1_m, r1_{d-m}), \tag{B.2}$$

and the post-test distribution is given by the covariance $\Sigma = \Sigma_0 + \Delta$. Notably, Our asymptotic framework generalizes to arbitrary covariance structures $\Gamma_0, \Delta$, and $A$, provided the empirical spectral distributions of these matrices converge weakly to probability measures on $\mathbb{R}_{\geq 0}$ with finite second moments. Under this Mean-Field regime, the macroscopic behavior of the learned operator $\widetilde{V}_*$ is determined by the spectral densities of the data and shift matrices, rather than their specific coordinate-level realizations.

**Decomposition of $\widetilde{V}_*$:** Starting from $\widetilde{V}_* = -\eta\Omega\Phi^\dagger - \Gamma_0^{-1}(I - \Pi_\Phi)$, with projection $\Pi_\Phi = \Phi\Phi^\dagger$.

Let $\Phi = M + \mathcal{E}$, where $M = A\Omega$ and $\mathcal{E}$ is the perturbation of $\Phi$ from its expectation $A\Omega$ with respect to randomness in the empirical features covariances $S_\tau$, for $\tau \in [B]$. Using the first-order expansion of the pseudoinverse:

$$\Pi_\Phi \approx (M + \mathcal{E})(M^\dagger - M^\dagger\mathcal{E}M^\dagger + \dots)$$

Multiplying this out and keeping only terms up to the first power of $\mathcal{E}$:

$$\Pi_\Phi \approx \underbrace{MM^\dagger}_{\Pi_\Omega} + \underbrace{\mathcal{E}M^\dagger - MM^\dagger\mathcal{E}M^\dagger}_{\text{First-order correction}}$$

We can simplify the correction term by factoring $MM^\dagger$:

$$\Pi_\Phi \approx MM^\dagger + (I - MM^\dagger)\mathcal{E}M^\dagger,$$
$$(I - \Pi_\Phi) \approx (I - MM^\dagger) - (I - MM^\dagger)\mathcal{E}M^\dagger.$$

By substituting this expanded projection back into the definition of $\widetilde{V}_*$, we get the following first-order approximation:

$$\widetilde{V}_* \approx -\eta\Omega(M^\dagger - M^\dagger\mathcal{E}M^\dagger) - \Gamma_0^{-1}\left[(I - MM^\dagger) - (I - MM^\dagger)\mathcal{E}M^\dagger\right] \tag{B.3}$$

Now, group the terms into deterministic ($V_S$) and stochastic ($V_N$) components. The Zero order component is given by:

$$V_S = -\eta\Omega M^\dagger - \Gamma_0^{-1}(I - MM^\dagger)$$

The first order component is given by:

$$V_N = -V_S\mathcal{E}M^\dagger$$

Equation (B.3) can be written as

$$\widetilde{V}_* \approx \widetilde{V} := V_S + V_N. \tag{B.4}$$

We next characterize the limit of test error using (5.1). For convenience we rewrite the characterization for the expected test error below:

$$\mathsf{Err}(\widetilde{V}) = \underbrace{\frac{1}{d}\,\mathbb{E}\big[\|I + \widetilde{V}\Sigma\|_F^2\big]}_{\text{Term I}} + \underbrace{\frac{1}{nd}\,\mathbb{E}\left[\mathrm{tr}(\widetilde{V}\Sigma^2\widetilde{V}^\mathsf{T}) + \mathrm{tr}(\widetilde{V}\Sigma\widetilde{V}^\mathsf{T})\mathrm{tr}(\Sigma)\right]}_{\text{Term II}} \tag{B.5}$$

where expectation is with respect to both the training and the test data. We also normalized the test error by the dimension $d$.

**Proportional regime.** We consider the proportional asymptotic regime, where $d, m, n, B \to \infty$, with $n$ the prompt length and $B$ the number of prompts. In addition, $B/d \to \beta$, $m/d \to \mu_1$, $d/n \to \gamma$ for some arbitrary but fixed constants $\beta, \mu_1, \gamma$. We also let $\mu_2 = 1 - \mu_1$.

**Notations.** The deterministic feature covariance $A$ is diagonal with block entries $a_1, a_2$. The test covariance $\Sigma$ is also diagonal with block diagonals $\Sigma_1, \Sigma_2$, namely

$$
\begin{aligned}
a_1 &:= \eta(\rho + 1), & a_2 &:= r \\
\Sigma_1 &:= \rho + 1, & \Sigma_2 &:= 1
\end{aligned}
\tag{B.6}
$$

Let $D_1 = I - \Gamma_0^{-1}\Sigma$ and $D_{pre} = \Gamma_0^{-1} - \eta A^{-1}$. Both are block-diagonal deterministic matrices and in the proportional asymptotic regime, we let $\alpha_k$ and $\tilde{\delta}_k$ be their respective scalar values on block $k \in \{1, 2\}$, and let $\delta_k = \tilde{\delta}_k \Sigma_k$. A simple calculation shows that with $\kappa := \gamma(\mu\rho + 1 - \mu)$, we have

$$
\begin{aligned}
\alpha_1 &= \frac{\kappa - 1}{\rho + \kappa}, & \alpha_2 &= \frac{\kappa}{\kappa + 1} \\
\delta_1 &= \frac{1 - \kappa}{\rho + \kappa}, & \tilde{\delta}_1 &= \frac{\delta_1}{\Sigma_1} = \frac{1 - \kappa}{(\rho + \kappa)(\rho + 1)} \\
\delta_2 &= \frac{1}{1 + \kappa} - \frac{\eta}{r}, & \tilde{\delta}_2 &= \frac{\delta_2}{\Sigma_2} = \delta_2
\end{aligned}
\tag{B.7}
$$

In addition, $-\Gamma_0^{-1}$ is also a block-diagonal deterministic matrix and in the asymptotic regime, we let $g_k$ be its respective scalar values on block $k \in \{1, 2\}$. It is easy to see that

$$
g_1 = -\frac{1}{\rho + \kappa}, \quad g_2 = -\frac{1}{1 + \kappa}
\tag{B.8}
$$

The matrix $\Sigma_A := \frac{1}{d}\left(\operatorname{tr}(A)A + A^2\right)$ is also block-diagonal and in the proportional regime, its respective scalar values on block $k \in \{1, 2\}$ converge to

$$
\begin{aligned}
s_1 &= (\mu_1 a_1 + \mu_2 a_2)a_1 = (\mu_1\eta(\rho+1) + \mu_2 r)\eta(\rho+1) \\
s_2 &= (\mu_1 a_1 + \mu_2 a_2)a_2 = (\mu_1\eta(\rho+1) + \mu_2 r)r
\end{aligned}
\tag{B.9}
$$

**Theorem B.1.** *Consider $\widetilde{V} = V_S + V_N$ the first order approximation of $\widetilde{V}_*$ as in (B.4). Under the proportional asymptotic regime, the following holds true:*

$$
\lim_{d\to\infty} \operatorname{Err}(\widetilde{V}) = Bias + \gamma \mathcal{T}_{inv}\mathcal{T}_{var} + \gamma\bar{\Sigma}\mathcal{T}_{var,\Sigma} + \gamma^2\bar{\Sigma}\mathcal{T}_{inv,\Sigma}\mathcal{T}_{var}
\tag{B.10}
$$

*where the terms are defined as follows, in terms of the notations defined by (B.6), (B.7), (B.8) and (B.9):*

- *Bias: For $\beta < 1$, let $q$ be the non-negative solution to:*

$$
\beta = \sum_{k=1}^{2} \mu_k \frac{a_k^2 q}{1 + a_k^2 q}
\tag{B.11}
$$

*and define $w_k = \frac{a_k^2 q}{1 + a_k^2 q}$, $v_k = w_k(1 - w_k)$, for $k \in \{1, 2\}$ and*

$$
T_{12} = \frac{\mu_1\mu_2 v_1 v_2}{\mu_1 v_1 + \mu_2 v_2}
\tag{B.12}
$$

*For $\beta \geq 1$, set $w_k = 1$, $v_k = 0$, and $T_{12} = 0$. We then have*

$$
Bias := \sum_{k=1}^{2} \mu_k \left[\alpha_k^2(1 - w_k) + (\alpha_k + \delta_k)^2 w_k\right] + T_{12}(\tilde{\delta}_2^2 - \tilde{\delta}_1^2)(\Sigma_1^2 - \Sigma_2^2)
\tag{B.13}
$$

- *The terms $\mathcal{T}_{inv}$ and $\mathcal{T}_{inv,\Sigma}$ are given by*

$$
\mathcal{T}_{inv} = \left\{ q\frac{\sum_{k=1}^{2} \mu_k \frac{\Sigma_k^2}{a_k^2} w_k^2}{\sum_{k=1}^{2} \mu_k \frac{1}{a_k^2} w_k^2} \right\} \mathbf{1}(\beta < 1) + \left\{ \frac{1}{\beta - 1}\sum_{k=1}^{2} \mu_k \frac{\Sigma_k^2}{a_k^2} \right\} \mathbf{1}(\beta > 1)
\tag{B.14}
$$

$$
\mathcal{T}_{inv,\Sigma} = \left\{ q\frac{\sum_{k=1}^{2} \mu_k \frac{\Sigma_k}{a_k^2} w_k^2}{\sum_{k=1}^{2} \mu_k \frac{1}{a_k^2} w_k^2} \right\} \mathbf{1}(\beta < 1) + \left\{ \frac{1}{\beta - 1}\sum_{k=1}^{2} \mu_k \frac{\Sigma_k}{a_k^2} \right\} \mathbf{1}(\beta > 1)
\tag{B.15}
$$

- *The terms $\mathcal{T}_{var,\mathcal{E}}$ and $\mathcal{T}_{var,\Sigma}$ are given by*

$$\mathcal{T}_{var} = \sum_{k=1}^{2} \mu_k s_k \left[ g_k^2 (1 - w_k) + (g_k + \tilde{\delta}_k)^2 w_k \right] - T_{12}(\tilde{\delta}_1 - \tilde{\delta}_2)(\tilde{\delta}_1 s_1 - \tilde{\delta}_2 s_2) \tag{B.16}$$

$$\mathcal{T}_{var,\Sigma} = \sum_{k=1}^{2} \mu_k \Sigma_k \left[ g_k^2 (1 - w_k) + (g_k + \tilde{\delta}_k)^2 w_k \right] - T_{12}(\tilde{\delta}_1 - \tilde{\delta}_2)(\tilde{\delta}_1 \Sigma_1 - \tilde{\delta}_2 \Sigma_2) \tag{B.17}$$

We next compare the predicted asymptotic limit of Err with numerical experiment. Recall $\widetilde{V}_*$ as the SFT loss minimizer given by (4.1), $\widetilde{V}$ its first order approximation, given by (B.4). In Figure 5 we plot $\mathsf{Err}(\widetilde{V}_*)$, $\mathsf{Err}(\widetilde{V})$ and our theoretical curve (B.10). As we see there is a great match between our theoretical prediction and simulation result for $(\mathsf{Err}(\widetilde{V}))$. In addition, it approximates $\mathsf{Err}(\widetilde{V}_*)$ reasonably well and the approximation becomes tighter as the prompt length $(n)$ grows (Figure 5b shows a better approximation at $n = 5000$ compared to Figure 5a for $n = 1000$).

Using Theorem B.10 we prove several properties of the asymptotic error and show that under interference, its minimum is achieved in the regime of $\beta < 1$. This confirms our Insight 2 in the main text, namely that SFT datasets should be curated to be relatively small in volume.

We denote the predicted theoretical error (right hand side of (B.10)) by $F(\beta)$, as function of $\beta$, as we would like to understand its behavior as $\beta$ varies.

**Proposition B.2.** *The followings hold true:*

$(i)$ $\lim_{\beta \to 1} F(\beta) = \infty$. *For $\beta > 1$, $F(\beta)$ is strictly decreasing. As $\beta \to \infty$, it converges to a finite asymptotic floor:*

$$F(\uparrow\infty) := \lim_{\beta \to \infty} F(\beta) = \sum_{k=1}^{2} \mu_k (\alpha_k + \delta_k)^2 + \gamma \bar{\Sigma} \sum_{k=1}^{2} \mu_k \Sigma_k (g_k + \tilde{\delta}_k)^2$$

$(ii)$ *We have*

$$F(0) = \sum_{k=1}^{2} \mu_k \alpha_k^2 + \gamma \bar{\Sigma} \sum_{k=1}^{2} \mu_k \Sigma_k g_k^2$$

*Also, $F(\uparrow\infty) - F(0) = C/r^2 + O(1/r)$, for a fixed $C > 0$. Consequently, for sufficiently small $r > 0$, $F(\uparrow\infty) > F(0)$. This guarantees that the global minimum of $F(\beta)$ is strictly achieved in the overparameterized regime ($\beta < 1$).*

$(iii)$ *Suppose that $\mu_1 \geq \frac{\rho^2}{1+\rho^2}$. For sufficiently small $r$ and $\gamma$, the initial derivative is strictly negative ($F'(0) < 0$). Hence, introducing a small number of prompts immediately and strictly decreases the test error.*

## C  Proof of Theorem B.1

### C.1  Analysis of Term I

We start by analyzing Term I. We have

$$\mathbb{E}\left[\left\| I + \widetilde{V}\Sigma \right\|_F^2\right] = \mathbb{E}\left[\left\| I + V_S\Sigma + V_N\Sigma \right\|_F^2\right] = \mathbb{E}\left[\left\| I + V_S\Sigma \right\|_F^2\right] + \mathbb{E}\left[\left\| V_N\Sigma \right\|_F^2\right]$$

because conditioned on $\Omega$, $\mathcal{E} = [(S_1 - A)w_1^*, \dots, (S_B - A)w_B^*]$ is zero mean and independent of $V_S$. Hence,

$$\lim_{d \to \infty} \frac{1}{d} \mathbb{E}\left[\left\| I + \widetilde{V}\Sigma \right\|_F^2\right] = \lim_{d \to \infty} \frac{1}{d} \mathbb{E}\left[\left\| I + V_S\Sigma \right\|_F^2\right] + \lim_{d \to \infty} \frac{1}{d} \mathbb{E}\left[\left\| V_N\Sigma \right\|_F^2\right]. \tag{C.1}$$

**Analysis of the Bias term.** The deterministic component of the test error (Bias) is governed by the matrix $M_S = I + V_S\Sigma$. We first express $V_S$ in terms of the orthogonal projection matrix $\Pi_M = MM^\dagger$. Using the identity $A^{-1}\Pi_M = \Omega M^\dagger$, we have:

$$V_S = -\eta A^{-1}\Pi_M - \Gamma_0^{-1}(I - \Pi_M) \tag{C.2}$$
$$M_S = (I - \Gamma_0^{-1}\Sigma) + (\Gamma_0^{-1} - \eta A^{-1})\Pi_M\Sigma \tag{C.3}$$

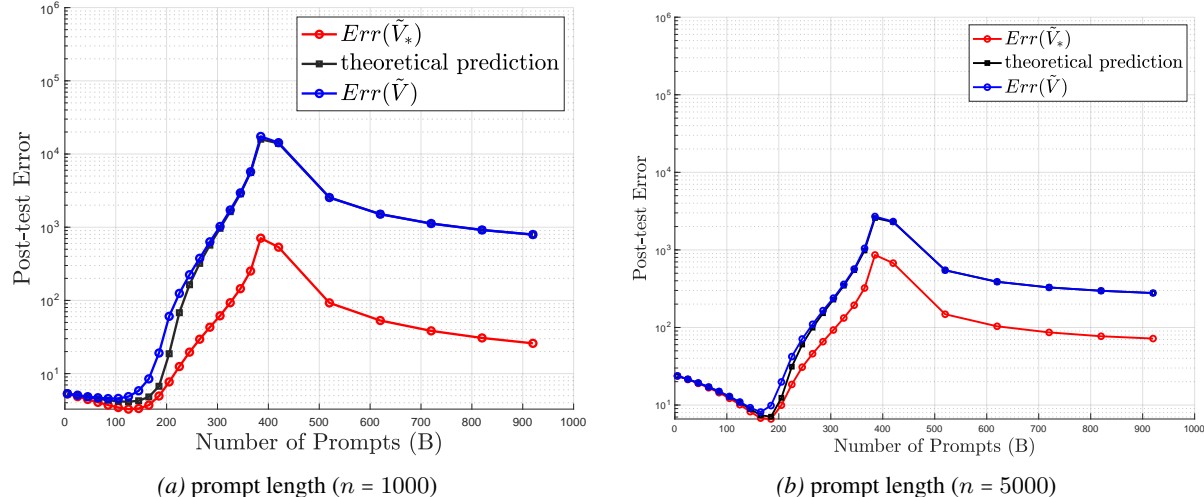

*(a)* prompt length $(n = 1000)$           *(b)* prompt length $(n = 5000)$

*Figure 5.* Comparison between theoretical prediction of the asymptotic error $\mathsf{Err}(\widetilde{V})$, the simulation results for $\mathsf{Err}(\widetilde{V})$ and $\mathsf{Err}(\widetilde{V}_*)$. We see a great match between theoretical prediction and simulation results. Here, $d = 600$, $m = 300$, $n = 600$ (prompt size), $\rho = 0.1$, $\eta = 0.2$, $r = 0.1$ (interference parameter). The simulations are averaged over 10 realizations.

Let $D_1 = I - \Gamma_0^{-1}\Sigma$ and $D_{pre} = \Gamma_0^{-1} - \eta A^{-1}$. Thus, $M_S = D_1 + D_{pre}\Pi_M\Sigma$. Note that $D_1$ and $D_{pre}$ are block-diagonal deterministic matrices, and in the asymptotic regime, their respective scalar values on block $k \in \{1, 2\}$ converge to $\alpha_k$ and $\tilde{\delta}_k$ given by (B.7). Let $\delta_k = \tilde{\delta}_k \Sigma_k$.

Expanding the normalized squared Frobenius norm, we obtain:

$$\frac{1}{d}\|M_S\|_F^2 = \frac{1}{d}\mathrm{tr}(D_1^2) + \frac{2}{d}\mathrm{tr}(D_1\Sigma\Pi_M D_{pre}) + \frac{1}{d}\mathrm{tr}(D_{pre}^2\Pi_M\Sigma^2\Pi_M) \tag{C.4}$$

We next note that

$$\frac{1}{d}\mathrm{tr}(D_1^2) = \sum_{k=1}^{2}\frac{d_k}{d}\alpha_k^2 = \sum_{k=1}^{2}\mu_k\alpha_k^2 \tag{C.5}$$

In addition, we have

$$\frac{2}{d}\mathrm{tr}(D_1\Sigma\Pi_M D_{pre}) = \frac{2}{d}\mathrm{tr}(D_{pre}D_1\Sigma\Pi_M) = \frac{2}{d}\sum_{k=1}^{2}\alpha_k\delta_k\mathrm{tr}(\Pi_{kk}) \tag{C.6}$$

To evaluate the quadratic trace $\mathrm{Quad} := \frac{1}{d}\mathrm{tr}(D_{pre}^2\Pi_M\Sigma^2\Pi_M)$, we partition the projection matrix into blocks $\Pi_{ij}$ with $i, j \in \{1, 2\}$ with $\Pi_{11}$ of size $m$ and $\Pi_{2,2}$ of size $d - m$. Let $T_{ij} = \frac{1}{d}\mathrm{tr}(\Pi_{ij}\Pi_{ji})$. Expanding the trace block-by-block yields:

$$\mathrm{Quad} = \delta_1^2 T_{11} + \delta_2^2 T_{22} + \left(\tilde{\delta}_2^2\Sigma_1^2 + \tilde{\delta}_1^2\Sigma_2^2\right)T_{12}. \tag{C.7}$$

Because $\Pi_M$ is a true orthogonal projection, $\Pi_M^2 = \Pi_M$. Examining the diagonal blocks of this identity gives

$$\Pi_{kk}^2 + \Pi_{kj}\Pi_{jk} = \Pi_{kk}, \tag{C.8}$$

with $k \neq j \in \{1, 2\}$.

Our next lemma characterizes the limit of normalized trace of $\Pi_{kk}$, using Stieltjes transform and Silverstein equation from the Random Matrix Theory.

**Lemma C.1.** *Let* $w_k := \lim_{d_k \to \infty}\frac{1}{d_k}\mathrm{tr}(\Pi_{kk})$. *Then, the following holds: For* $\beta < 1$,

$$w_k = \frac{a_k^2 q}{1 + a_k^2 q}, \tag{C.9}$$

*with q being the non-negative solution to:*

$$\beta = \sum_{k=1}^{2} \mu_k \frac{a_k^2 q}{1 + a_k^2 q} \tag{C.10}$$

*For $\beta \geq 1$, we have $w_k = 1$.*

Taking the normalized trace from (C.8) gives:

$$T_{kk} = \mu_k w_k - T_{12}, \quad k \in \{1, 2\} \tag{C.11}$$

Substituting this into the quadratic term and simplifying:

$$\text{Quad} = \mu_1 \delta_1^2 w_1 + \mu_2 \delta_2^2 w_2 + T_{12}(\tilde{\delta}_2^2 - \tilde{\delta}_1^2)(\Sigma_1^2 - \Sigma_2^2) \tag{C.12}$$

Also by recalling (C.6) we have

$$\lim_{d \to \infty} \frac{2}{d} \text{tr}(D_1 \Sigma \Pi_M D_{pre}) = \frac{2}{d} \text{tr}(D_1 \Sigma \Pi_M D_{pre}) = \lim_{d \to \infty} 2 \sum_{k=1}^{2} \left(\frac{d_k}{d}\right) \alpha_k \delta_k \left(\frac{1}{d_k} \text{tr}(\Pi_{kk})\right) = 2 \sum_{k=1}^{2} \mu_k \alpha_k \delta_k w_k \tag{C.13}$$

Combining the linear and quadratic traces given by (C.5), (C.13) and (C.12), the complete rigorous bias evaluates to:

$$\lim_{d \to \infty} \frac{1}{d} \|M_S\|_F^2 = \sum_{k=1}^{2} \mu_k \left[\alpha_k^2(1 - w_k) + (\alpha_k + \delta_k)^2 w_k\right] + T_{12}(\tilde{\delta}_2^2 - \tilde{\delta}_1^2)(\Sigma_1^2 - \Sigma_2^2). \tag{C.14}$$

In the next lemma, we characterize $T_{12}$ which completes our analysis of the Bias term.

**Lemma C.2.** *Let $\Pi_M$ be the orthogonal projection matrix onto the column space of $M = A\Omega$, where $\Omega \in \mathbb{R}^{d \times B}$ has i.i.d. entries of variance $1/d$, and $A$ is a deterministic block-diagonal matrix with block dimensions $d_k = \mu_k d$ and corresponding squared eigenvalues $a_k^2$ for $k \in \{1, 2\}$. Let $\Pi_{ij}$ denote the sub-blocks of $\Pi_M$. As $d, B \to \infty$ with $B/d \to \beta$, the normalized cross-subspace leakage trace $T_{12} = \lim_{d \to \infty} \frac{1}{d} \text{tr}(\Pi_{12}\Pi_{21})$ is almost surely given by:*

$$T_{12} = \frac{\mu_1 \mu_2 v_1 v_2}{\mu_1 v_1 + \mu_2 v_2} \tag{C.15}$$

*where $v_k = w_k(1 - w_k)$ is the variance factor of the projection on block $k$, and $w_k = \frac{a_k^2 q}{1 + a_k^2 q}$ are the Stieltjes weights defined by the fixed-point root $q$.*

**Analysis of the noise term.** We recall the dimension ratios as $\mu_1 = m/d$ and $\gamma = d/n$. The noise operator acting on the test covariance is defined exactly as $V_N \Sigma = -V_S \mathcal{E} M^\dagger \Sigma$. We seek the limit of the normalized expected squared Frobenius norm:

$$\frac{1}{d} \mathbb{E}_{\mathcal{E}} \left[\|V_N \Sigma\|_F^2\right] = \frac{1}{d} \mathbb{E}_{\mathcal{E}} \left[\text{tr}\left(V_S \mathcal{E} M^\dagger \Sigma^2 (M^\dagger)^T \mathcal{E}^T V_S^T\right)\right] \tag{C.16}$$

Let $Q = M^\dagger \Sigma^2 (M^\dagger)^T$. Because $M = \mathbb{E}[\Phi]$ is deterministic, $Q$ is constant with respect to the noise realization $\mathcal{E}$.

Let $\varepsilon_\tau = (S_\tau - A)w_\tau^*$ be the $\tau$-th column of $\mathcal{E}$. We first compute the expectation over the feature samples $x_{i,\tau}$ conditioned on the weight matrix $\Omega$. Since the feature samples are independent across different weights $\tau$, the columns of $\mathcal{E}$ are mutually independent with zero mean:

$$\mathbb{E}[\varepsilon_\tau \varepsilon_\gamma^\top \mid \Omega] = 0 \quad \text{for } \tau \neq \gamma$$

For the diagonal terms, we use the standard identity for the covariance of a Wishart quadratic form. For any fixed vector $u$ and $S \sim W_d(n, \frac{1}{n}A)$:

$$\mathbb{E}[(S - A)uu^\top(S - A)] = \frac{1}{n}\left((u^\top A u)A + Auu^\top A\right)$$

Summing over the entries of $Q$:

$$\mathbb{E}_{\mathcal{E}|\Omega}[\mathcal{E}Q\mathcal{E}^\top] = \sum_{\tau \neq \gamma} Q_{\tau\gamma} \mathbb{E}[\varepsilon_\tau \varepsilon_\gamma^\top \mid \Omega] = \frac{1}{n} \sum_{\tau=1}^{B} Q_{\tau\tau}\left((w_\tau^{*\top} A w_\tau^*)A + A w_\tau^* w_\tau^{*\top} A\right)$$

We now take the expectation over $\Omega$. By the rotational invariance of the Gaussian distribution, the expectation of any function of $\Omega$ that is equivariant under orthogonal transformations must be isotropic. In particular, the term $Z = \mathbb{E}\left[\sum_\tau Q_{\tau\tau} w_\tau^* w_\tau^{*\top}\right]$ must satisfy $Z = cI_d$. Taking the trace:

$$cd = \mathbb{E}\left[\sum_{\tau=1}^{B} Q_{\tau\tau}\|w_\tau^*\|^2\right] = d\mathbb{E}[\mathrm{tr}(Q)] \implies c = \mathbb{E}[\mathrm{tr}(Q)]$$

In the high-dimensional limit, the correlation between the weight-norm quadratic form $(w_\tau^{*\top} A w_\tau^*)$ and the kernel diagonal $Q_{\tau\tau}$ vanishes, by which we obtain

$$\mathbb{E}\left[\sum_{\tau=1}^{B} Q_{\tau\tau}(w_\tau^{*\top} A w_\tau^*)\right] = \mathbb{E}[\mathrm{tr}(Q)]\mathrm{tr}(A)$$

Combining these, we obtain:

$$\mathbb{E}[\mathcal{E}Q\mathcal{E}^\top] = \frac{\mathbb{E}[\mathrm{tr}(Q)]}{n}\left(\mathrm{tr}(A)A + A^2\right) \tag{C.17}$$

We set $\Sigma_A := \frac{1}{d}\left(\mathrm{tr}(A)A + A^2\right)$. Substituting the above identity into (C.16) we obtain:

$$\frac{1}{d}\mathbb{E}_\mathcal{E}\left[\|V_N\Sigma\|_F^2\right] = \frac{1}{d}\mathrm{tr}\left(V_S\left[\gamma\mathrm{tr}(Q)\Sigma_A\right]V_S^T\right) \tag{C.18}$$

Because $\mathrm{tr}(Q)$ is a scalar, it factors entirely out of the matrix product. Using the property $(M^\dagger)^T M^\dagger = (MM^T)^\dagger$, the expectation rigorously splits into the product of two independent, normalized trace functionals:

$$\frac{1}{d}\mathbb{E}[\|V_N\Sigma\|_F^2] = \gamma\underbrace{\left[\mathrm{tr}(\Sigma^2(MM^T)^\dagger)\right]}_{\mathcal{T}_{inv}}\cdot\underbrace{\left[\frac{1}{d}\mathrm{tr}(V_S\Sigma_A V_S^T)\right]}_{\mathcal{T}_{var}} \tag{C.19}$$

**Derivation of the pseudo-inverse trace** ($\mathcal{T}_{inv}$)**.** We must evaluate the target trace $\mathcal{T}_{inv} = \mathrm{tr}(\Sigma^2(MM^T)^\dagger)$, which governs the variance functional. Because the feature matrix $M = A\Omega$ is constructed with $\Omega \sim \mathcal{N}(0,1)$ i.i.d. entries, the unscaled matrix $G = MM^T$ has eigenvalues scaling as $O(d)$. To rigorously apply the Stieltjes transform, we define the normalized matrix $\hat{G} = \frac{1}{d}G$, which has $O(1)$ eigenvalues. The target trace scales as:

$$\mathcal{T}_{inv} = \mathrm{tr}\left(\Sigma^2(d\hat{G})^\dagger\right) = \frac{1}{d}\mathrm{tr}\left(\Sigma^2\hat{G}^\dagger\right) \tag{C.20}$$

• **Over-parameterized regime** ($\beta < 1$)**.** Because $\hat{G}$ is strictly singular in the over-parameterized regime ($B < d$), direct inversion is invalid. To evaluate the trace rigorously for any aspect ratio $\beta$, we introduce a strictly positive regularization parameter $z > 0$ and define the perturbed resolvent:

$$R(z,t) = (\hat{G} + tA\Sigma^2 A + zI_d)^{-1} \tag{C.21}$$

Let $m(z,t) = \frac{1}{d}\mathrm{tr}\left(R(z,t)\right)$ be its normalized trace. Because $z > 0$, $R(z,t)$ is unconditionally invertible and bounded for all $B$. Taking the derivative of $m(z,t)$ with respect to the continuous perturbation $t$ at $t = 0$ yields:

$$\frac{\partial}{\partial t}m(z,t)\Big|_{t=0} = -\frac{1}{d}\mathrm{tr}\left(R(z,0)(A\Sigma^2 A)R(z,0)\right) \tag{C.22}$$

Because $R(z,0)$, $A$, and $\Sigma$ are all well-defined, finite $d \times d$ matrices, we can validly apply cyclic permutation to the trace. We move the rightmost $R(z,0)$ to the left, and use the fact that the diagonal matrices $A$ and $\Sigma^2$ commute ($A\Sigma^2 A = \Sigma^2 A^2$):

$$\frac{\partial}{\partial t}m(z,t)\Big|_{t=0} = -\frac{1}{d}\mathrm{tr}\left((A\Sigma^2 A)R(z,0)^2\right) = -\frac{1}{d}\mathrm{tr}\left(\Sigma^2 A^2 R(z,0)^2\right) \tag{C.23}$$

We now define our target Stieltjes derivative $q'(0)$ as the limit of this regularized derivative as $z \to 0^+$. By defining the operator limit $\lim_{z\to 0^+} R(z,0)^2 \equiv (\hat{G}^\dagger)^2$ strictly on the non-null subspace, this identically maps to our target variance trace $\mathcal{T}_{inv}$ across all parameterization regimes:

$$-q'(0) = \lim_{z\to 0^+} \frac{1}{d}\mathrm{tr}\big(\Sigma^2 A^2 R(z,0)^2\big) \equiv \frac{1}{d}\mathrm{tr}\big(\Sigma^2 A^2 (\hat{G}^\dagger)^2\big) = \mathcal{T}_{inv} \tag{C.24}$$

To find $q'(0)$ analytically, we differentiate the fixed-point equation of the perturbed resolvent. The eigenvalues of the perturbed deterministic envelope are $a_k^2(1+t\Sigma_k^2)$. Using the Silverstein equation, we have the following fixed-point equation:

$$\beta = \sum_{k=1}^{2} \mu_k \frac{a_k^2(1+t\Sigma_k^2)q(t)}{1+a_k^2(1+t\Sigma_k^2)q(t)} \tag{C.25}$$

Differentiating both sides with respect to $t$ at $t=0$ (where $q(0) = q$) gives:

$$0 = \sum_{k=1}^{2} \mu_k \frac{a_k^2\Sigma_k^2 q + a_k^2 q'(0)}{(1+a_k^2 q)^2} \tag{C.26}$$

Separating the terms and recognizing that the effective block weights are $w_k = \frac{a_k^2 q}{1+a_k^2 q}$, we observe the algebraic identity $\frac{a_k^2}{(1+a_k^2 q)^2} = \frac{w_k^2}{a_k^2 q^2}$. Substituting this into the differential equation gives:

$$-q'(0) \sum_{k=1}^{2} \mu_k \frac{w_k^2}{a_k^2 q^2} = q \sum_{k=1}^{2} \mu_k \Sigma_k^2 \frac{w_k^2}{a_k^2 q^2} \tag{C.27}$$

Multiplying by $q^2$ and isolating $-q'(0)$, we obtain the exact closed-form limit:

$$\mathcal{T}_{inv} = q \frac{\sum_{k=1}^{2} \mu_k \frac{\Sigma_k^2}{a_k^2} w_k^2}{\sum_{k=1}^{2} \mu_k \frac{1}{a_k^2} w_k^2} \tag{C.28}$$

• **Under-parameterized regime** ($\beta > 1$). The differential Stieltjes approach relies on the fixed-point root $q$ being finite, which holds strictly for the over-parameterized regime ($\beta < 1$). When $\beta > 1$, the number of samples exceeds the ambient dimension ($B > d$), causing the rank fraction to saturate at $\bar{\beta} = 1$, which mathematically drives $q \to \infty$.

However, in this over-parameterized regime, the unscaled feature covariance matrix $G = MM^T$ becomes strictly full rank almost surely. Consequently, the normalized matrix $\hat{G} = \frac{1}{d}G$ is strictly invertible, and its pseudoinverse reduces to the standard inverse $\hat{G}^{-1}$. We skip the perturbation derivative and evaluate the trace directly using the deterministic equivalent for the inverse of a generalized sample covariance matrix. Note that $\hat{G}^{-1} = A^{-1}W^{-1}A^{-1}$ with $W = \frac{1}{d}\Omega\Omega^T$ a standard Wishart matrix of size $d \times B$ and so by the inverse moments of the Marchenko-Pastur law, its deterministic equivalent is given by $W \asymp \frac{1}{\beta-1}I_d$, which implies that

$$\hat{G}^{-1} \asymp \frac{1}{\beta-1}(A^2)^{-1} \tag{C.29}$$

Substituting this deterministic equivalent directly into the target trace functional yields the exact closed-form limit for $\beta > 1$:

$$\mathcal{T}_{inv} = \frac{1}{d}\mathrm{tr}\left(\Sigma^2\left[\frac{1}{\beta-1}A^{-2}\right]\right) = \frac{1}{\beta-1}\sum_{k=1}^{2}\mu_k \frac{\Sigma_k^2}{a_k^2} \tag{C.30}$$

Equations (C.28) and (C.30) both diverge at the interpolation threshold ($\beta = 1$).

We combine both equation into one unifying relation:

$$\mathcal{T}_{inv} = \left\{ q\frac{\sum_{k=1}^{2}\mu_k \frac{\Sigma_k^2}{a_k^2}w_k^2}{\sum_{k=1}^{2}\mu_k \frac{1}{a_k^2}w_k^2} \right\}\mathbf{1}(\beta < 1) + \left\{\frac{1}{\beta-1}\sum_{k=1}^{2}\mu_k \frac{\Sigma_k^2}{a_k^2}\right\}\mathbf{1}(\beta > 1) \tag{C.31}$$

**Derivation of the trace term** ($\mathcal{T}_{var}$)**.** We evaluate the trace term $\mathcal{T}_{var} = \lim_{d\to\infty} \frac{1}{d}\text{tr}(V_S \Sigma_A V_S^T)$. Recall the deterministic test operator $V_S = -\Gamma_0^{-1} + D_{pre}\Pi_M$, where $D_{pre} = \Gamma_0^{-1} - \eta A^{-1}$. Note that $D_{pre}$ and $-\Gamma_0^{-1}$ are both block-diagonal deterministic matrices. Also their respective scalar values on block $k \in \{1, 2\}$ in the proportional asymptotic regime converges to $\tilde{\delta}_k$ and $g_k$ given by (B.7) and (B.8). Expanding the trace yields:

$$\mathcal{T}_{var} = \frac{1}{d}\text{tr}(\Gamma_0^{-2}\Sigma_A) + \frac{2}{d}\text{tr}(-\Gamma_0^{-1}\Sigma_A D_{pre}\Pi_M) + \frac{1}{d}\text{tr}(D_{pre}\Pi_M \Sigma_A D_{pre}\Pi_M) \tag{C.32}$$

Let $T_{ij} = \frac{1}{d}\text{tr}(\Pi_{ij}\Pi_{ji})$. The linear traces evaluate strictly on the diagonal blocks. Similar to derivations (C.5) and (C.13) we have

$$\lim_{d\to\infty} \text{tr}(\Gamma_0^{-2}\Sigma_A) = \sum_{k=1}^{2} \mu_k g_k^2 s_k$$

where $s_1$ and $s_2$ are the limit of the scalar on the blocks of $\Sigma_A$ given by (B.9). In addition,

$$\lim_{d\to\infty} \frac{2}{d}\text{tr}(-\Gamma_0^{-1}\Sigma_A D_{pre}\Pi_M) = 2\sum_{k=1}^{2} \mu_k g_k \tilde{\delta}_k s_k w_k$$

The quadratic trace $\text{Quad} := \frac{1}{d}\text{tr}(D_{pre}\Pi_M \Sigma_A D_{pre}\Pi_M)$ expands over the $2 \times 2$ block partition as:

$$\text{Quad} = \tilde{\delta}_1^2 s_1 T_{11} + \tilde{\delta}_2^2 s_2 T_{22} + \tilde{\delta}_1\tilde{\delta}_2(s_1 + s_2)T_{12} \tag{C.33}$$

Invoking (C.11), we have $T_{kk} = \mu_k w_k - T_{12}$. Substituting these constraints into the quadratic expansion yields:

$$\text{Quad} = \sum_{k=1}^{2} \mu_k \tilde{\delta}_k^2 s_k w_k - T_{12}\left[\tilde{\delta}_1^2 s_1 + \tilde{\delta}_2^2 s_2 - \tilde{\delta}_1\tilde{\delta}_2 s_1 - \tilde{\delta}_1\tilde{\delta}_2 s_2\right] \tag{C.34}$$

The bracketed multiplier for $T_{12}$ factors analytically into $(\tilde{\delta}_1 - \tilde{\delta}_2)(\tilde{\delta}_1 s_1 - \tilde{\delta}_2 s_2)$. Recombining the linear and quadratic components completes the square for the diagonal elements, yielding:

$$\mathcal{T}_{var} = \sum_{k=1}^{2} \mu_k s_k \left[g_k^2(1 - w_k) + (g_k + \tilde{\delta}_k)^2 w_k\right] - T_{12}(\tilde{\delta}_1 - \tilde{\delta}_2)(\tilde{\delta}_1 s_1 - \tilde{\delta}_2 s_2) \tag{C.35}$$

By recalling (C.19), the noise limit $\frac{1}{d}\mathbb{E}_{\mathcal{E}}\left[\|V_N\Sigma\|_F^2\right]$ is given by the product of equations (C.28) and (C.35).

## C.2  Analysis of Term II

Since $\text{tr}(\Sigma)$ scales as $O(d)$, the $\text{tr}(\widetilde{V}\Sigma\widetilde{V}^T)\text{tr}(\Sigma)$ term dominates the $\text{tr}(\widetilde{V}\Sigma^2\widetilde{V}^T)$ term in the high-dimensional limit. Letting $\bar{\Sigma} = \lim_{d\to\infty} \frac{1}{d}\text{tr}(\Sigma)$, the dominant component of Term II evaluates to:

$$\text{Term II} = \gamma\bar{\Sigma} \cdot \frac{1}{d}\mathbb{E}\left[\text{tr}(\tilde{V}\Sigma\tilde{V}^T)\right] \tag{C.36}$$

Recall that $\tilde{V} = V_S + V_N$ with $V_N$ zero mean. Since the cross-terms are zero, we get the following decomposition:

$$\text{Term II} = \underbrace{\gamma\bar{\Sigma}\frac{1}{d}\text{tr}(V_S\Sigma V_S^T)}_{\text{Term II Signal}} + \underbrace{\gamma\bar{\Sigma}\frac{1}{d}\mathbb{E}_{\mathcal{E}}\left[\text{tr}(V_N\Sigma V_N^T)\right]}_{\text{Term II Noise}} \tag{C.37}$$

**Derivation of Term II Signal.** We evaluate $\mathcal{T}_{var,\Sigma} = \frac{1}{d}\text{tr}(V_S\Sigma V_S^T)$. The deterministic operator is $V_S = -\Gamma_0^{-1} + D_{pre}\Pi_M$, where $D_{pre} = \Gamma_0^{-1} - \eta A^{-1}$ are block-diagonal. Expanding the trace yields:

$$\mathcal{T}_{var,\Sigma} = \frac{1}{d}\text{tr}(\Gamma_0^{-2}\Sigma) + \frac{2}{d}\text{tr}(-\Gamma_0^{-1}\Sigma D_{pre}\Pi_M) + \frac{1}{d}\text{tr}(D_{pre}\Pi_M\Sigma D_{pre}\Pi_M) \tag{C.38}$$

As we observe the expression for $\mathcal{T}_{var,\Sigma}$ is same as $\mathcal{T}_{var}$ with $\Sigma_A$ replaced by $\Sigma$. Hence, by a similar derivation of (C.35) we get

$$\mathcal{T}_{var,\Sigma} = \sum_{k=1}^{2} \mu_k \Sigma_k \left[ g_k^2 (1 - w_k) + (g_k + \tilde{\delta}_k)^2 w_k \right] - T_{12}(\tilde{\delta}_1 - \tilde{\delta}_2)(\tilde{\delta}_1 \Sigma_1 - \tilde{\delta}_2 \Sigma_2) \tag{C.39}$$

**Derivation of Term II Noise.** We must evaluate $\frac{1}{d}\mathbb{E}\left[\text{tr}(V_N \Sigma V_N^T)\right]$. Following similar derivation of (C.19), replacing $\Sigma$ by $\Sigma^{1/2}$, we arrive at

$$\frac{1}{d}\mathbb{E}[\|V_N \Sigma^{1/2}\|_F^2] = \gamma \underbrace{\left[\text{tr}(\Sigma(MM^T)^\dagger)\right]}_{\mathcal{T}_{inv,\Sigma}} \cdot \underbrace{\left[\frac{1}{d}\text{tr}(V_S \hat{\Sigma}_\varepsilon V_S^T)\right]}_{\mathcal{T}_{var}} \tag{C.40}$$

Notice that we already characterized $\mathcal{T}_{var}$ in the analysis of Term I.

We next evaluate $\mathcal{T}_{inv,\Sigma} = \lim_{d\to\infty} \frac{1}{d}\text{tr}(\Sigma G^\dagger)$, where $G = MM^\dagger = A\Omega\Omega^T A$. Note that the expression for $\mathcal{T}_{inv,\Sigma}$ is same as $\mathcal{T}_{inv}$ where $\Sigma^2$ is replaced by $\Sigma$. Following the same derivation for (C.31), we arrive at

$$\mathcal{T}_{inv,\Sigma} = \left\{ q\frac{\sum_{k=1}^{2} \mu_k \frac{\Sigma_k}{a_k^2} w_k^2}{\sum_{k=1}^{2} \mu_k \frac{1}{a_k^2} w_k^2} \right\} \mathbf{1}(\beta < 1) + \left\{ \frac{1}{\beta - 1} \sum_{k=1}^{2} \mu_k \frac{\Sigma_k}{a_k^2} \right\} \mathbf{1}(\beta > 1) \tag{C.41}$$

Combining the above characterizations, the limit for the components of Term II are given by:

$$\text{Term II Signal} = \gamma\bar{\Sigma} \cdot \mathcal{T}_{var,\Sigma} \tag{C.42}$$

$$\text{Term II Noise} = \gamma^2\bar{\Sigma} \cdot \mathcal{T}_{inv,\Sigma} \cdot \mathcal{T}_{var} \tag{C.43}$$

where $\mathcal{T}_{var,\Sigma}$ is given by Eq. (C.39), $\mathcal{T}_{inv,\Sigma}$ by Eq. (C.41), and $\mathcal{T}_{var}$ is given by (C.35) from the Term I derivation. Putting the characterizations derived for Term I and Term II in (B.5) completes the proof.

### C.2.1 PROOF OF LEMMA C.1

To evaluate the asymptotic trace of $\Pi_M$, we express the orthogonal projection operator onto the column space of the empirical feature matrix $M$ as the limit of a Ridge-regularized inverse as the regularization parameter $z \to 0^+$:

$$\Pi_M = \lim_{z\to 0^+} M(M^T M + zI_B)^{-1}M^T = I_d - \lim_{z\to 0^+} z(G + zI_d)^{-1} \tag{C.44}$$

where $G = MM^T = A\Omega\Omega^T A$ is the generalized sample covariance matrix, and $R(z) = (G + zI_d)^{-1}$ is its resolvent.

By the Bai-Silverstein theorem, as $d, B \to \infty$ with $B/d \to \beta$, the random resolvent $R(z)$ is asymptotically equivalent to a deterministic diagonal matrix $T(z)$. For any bounded deterministic matrix $D$, the normalized trace converges almost surely:

$$\lim_{d\to\infty} \frac{1}{d}\text{tr}(DR(z)) - \frac{1}{d}\text{tr}(DT(z)) \xrightarrow{a.s.} 0 \tag{C.45}$$

where $T(z)$ is given by

$$T(z) = \left(zI_d + v(z)A^2\right)^{-1},$$

and $v(z)$ is the Stieltjes transform of the companion matrix $\tilde{G} = \Omega^T A^2 \Omega$.

We define the effective rank fraction preserved in the $k$-th block as the normalized trace of the projection matrix restricted to that subspace:

$$w_k = \lim_{d\to\infty} \frac{1}{d_k}\text{tr}(\Pi_{kk}) \tag{C.46}$$

Substituting the resolvent limit and its deterministic equivalent $T(z)$:

$$
\begin{aligned}
w_k &= 1 - \lim_{z \to 0^+} \frac{1}{d_k} \sum_{i \in \text{Block } k} z T_{ii}(z) \\
&= 1 - \lim_{z \to 0^+} \frac{z}{z + a_k^2 v(z)} \\
&= 1 - \lim_{z \to 0^+} \frac{1}{1 + a_k^2 \frac{v(z)}{z}}
\end{aligned}
\tag{C.47}
$$

We define the strict Stieltjes fixed-point root $q$ as the limit of this ratio near the origin:

$$
q = \lim_{z \to 0^+} \frac{v(z)}{z}
\tag{C.48}
$$

Substituting $q$ into the limit yields the following relation for the block weights:

$$
w_k = 1 - \frac{1}{1 + a_k^2 q} = \frac{a_k^2 q}{1 + a_k^2 q}
\tag{C.49}
$$

To determine the fixed-point root $q$, we utilize the trace identity between the resolvents of the $d \times d$ generalized sample covariance matrix $G$ and its $B \times B$ companion matrix $\tilde{G} = \Omega^T A^2 \Omega$. Because their non-zero eigenvalues are strictly identical, the normalized trace of the feature resolvent, $m(z) = \frac{1}{d}\text{tr}(R(z)) = \frac{1}{d}\text{tr}[(G + z I_d)^{-1}]$, is given by:

$$
z m(z) = 1 - \beta + \beta z v(z).
\tag{C.50}
$$

By the Bai-Silverstein theorem, $m(z)$ is asymptotically equivalent to the trace of the deterministic matrix $T(z)$. Substituting this deterministic equivalent yields:

$$
m(z) = \sum_{k=1}^{K} \mu_k \frac{1}{z + a_k^2 v(z)}
\tag{C.51}
$$

Multiplying by $z$ and equating this to the trace identity (C.50) establishes the exact relation:

$$
1 - \beta + \beta z v(z) = \sum_{k=1}^{K} \mu_k \frac{z}{z + a_k^2 v(z)} = \sum_{k=1}^{K} \mu_k \frac{1}{1 + a_k^2 \frac{v(z)}{z}}
\tag{C.52}
$$

We evaluate the strict limit of this equation as $z \to 0^+$. On the right side, we substitute our definition of the root $q = \lim_{z \to 0^+} \frac{v(z)}{z}$. On the left side, the limit of $z v(z)$ is governed by the dimension of the null space of the companion matrix $\tilde{G}$. The maximum rank of $\tilde{G}$ is bounded by $d$. If $B > d$ (i.e., $\beta > 1$), the companion matrix has exactly $B - d$ strict zero eigenvalues and the resolvent trace scales proportionally to $\frac{B-d}{B}\frac{1}{z}$. We therefore evaluate the limit exactly as:

$$
\lim_{z \to 0^+} z v(z) = \max\left(1 - \frac{1}{\beta}, 0\right)
\tag{C.53}
$$

Substituting these limits into both sides of the trace identity yields:

$$
1 - \beta + \beta \max\left(1 - \frac{1}{\beta}, 0\right) = \sum_{k=1}^{K} \mu_k \frac{1}{1 + a_k^2 q}
\tag{C.54}
$$

The left side mathematically simplifies exactly to $\max(1 - \beta, 0)$. On the right side, we substitute the definition of the block weights $w_k = \frac{a_k^2 q}{1 + a_k^2 q}$, utilizing the identity $\frac{1}{1 + a_k^2 q} = 1 - w_k$:

$$
\max(1 - \beta, 0) = \sum_{k=1}^{K} \mu_k (1 - w_k) = 1 - \sum_{k=1}^{K} \mu_k w_k
\tag{C.55}
$$

Rearranging the terms immediately yields:

$$
\sum_{k=1}^{K} \mu_k w_k = 1 - \max(1 - \beta, 0) = \min(\beta, 1) = \bar{\beta}
\tag{C.56}
$$

This derivation holds universally across all parameterization regimes. In the under-parameterized regime ($\beta > 1$), the effective rank fraction saturates at $\bar{\beta} = 1$, which mathematically forces $w_k \to 1$ and $q \to \infty$. Equations (C.49) and (C.56) completely and deterministically parameterize the finite-dimensional traces of the random projection $\Pi_M$.

### C.2.2 PROOF OF LEMMA C.2

We express the orthogonal projection matrix $\Pi_M$ as the limit of the regularized resolvent $R(z) = (A\Omega\Omega^T A + zI_d)^{-1}$ as $z \to 0^+$:

$$\Pi_M = I_d - \lim_{z \to 0^+} zR(z) \tag{C.57}$$

Let $D_1$ and $D_2$ be the orthogonal block indicator matrices for subspaces 1 and 2, such that $D_1 D_2 = 0$. Specifically,

$$D_1 = \begin{bmatrix} I_m & 0_{m \times (d-m)} \\ 0_{(d-m) \times m} & 0_{d-m} \end{bmatrix}, \quad D_2 = \begin{bmatrix} 0_m & 0_{m \times (d-m)} \\ 0_{(d-m) \times m} & I_{d-m} \end{bmatrix} \tag{C.58}$$

The cross-trace can be written as $\mathrm{tr}(\Pi_{12}\Pi_{21}) = \mathrm{tr}(D_1 \Pi_M D_2 \Pi_M)$. Substituting the resolvent limit into the trace definition yields:

$$T_{12} = \lim_{z \to 0^+} \lim_{d \to \infty} \frac{1}{d} \mathrm{tr}\left(D_1(I_d - zR(z))D_2(I_d - zR(z))\right) \tag{C.59}$$

Because $D_1 D_2 = 0$, expanding the product causes all terms of order lower than $R(z)^2$ to vanish exactly:

$$T_{12} = \lim_{z \to 0^+} z^2 \left[ \lim_{d \to \infty} \frac{1}{d} \mathrm{tr}\left(D_1 R(z) D_2 R(z)\right) \right] \tag{C.60}$$

In the next lemma, we characterize the inner limit.

**Lemma C.3.** *Let $R(z) = (A\Omega\Omega^T A + zI_d)^{-1}$ be the resolvent of the generalized sample covariance matrix, and let $T(z) = (zI_d + v(z)A^2)^{-1}$ be its deterministic equivalent. Let $D_1$ and $D_2$ be $d \times d$ diagonal orthogonal block indicator matrices such that $D_1 D_2 = 0$. In the high-dimensional limit $d, B \to \infty$ with $B/d \to \beta$, the normalized trace of the product of the two resolvents converges almost surely to:*

$$\lim_{d \to \infty} \frac{1}{d} tr\left(D_1 R(z) D_2 R(z)\right) = \frac{v(z)^2}{\beta} \frac{\Psi_1(z)\Psi_2(z)}{\Delta(z)} \tag{C.61}$$

*where $\Psi_k$ and $\Delta_k$ are defined as:*

$$\Psi_k(z) = \lim_{d \to \infty} \frac{1}{d} tr\left(D_k A^2 T(z)^2\right),$$

$$\Delta(z) = 1 - \frac{v(z)^2}{\beta} \lim_{d \to \infty} \frac{1}{d} tr\left(A^4 T(z)^2\right).$$

Using Lemma C.3, we have

$$\Psi_k(z) = \lim_{d \to \infty} \frac{1}{d} \mathrm{tr}\left(D_k T(z) A^2 T(z)\right) = \frac{\mu_k a_k^2}{(z + a_k^2 v(z))^2} \tag{C.62}$$

$$\Delta(z) = 1 - \frac{v(z)^2}{\beta} \lim_{d \to \infty} \frac{1}{d} \mathrm{tr}\left(A^4 T(z)^2\right) = 1 - \frac{v(z)^2}{\beta} \sum_{k=1}^{2} \frac{\mu_k a_k^4}{(z + a_k^2 v(z))^2} \tag{C.63}$$

We now evaluate the limit as $z \to 0^+$. Using the Stieltjes fixed-point definition $q = \lim_{z \to 0^+} \frac{v(z)}{z}$, we have $v(z) = qz + o(z)$. First, we evaluate the limit of the scaled block traces $z^2 \Psi_k(z)$:

$$\lim_{z \to 0^+} z^2 \Psi_k(z) = \lim_{z \to 0^+} \frac{z^2 \mu_k a_k^2}{z^2(1 + a_k^2 q)^2} = \frac{\mu_k a_k^2}{(1 + a_k^2 q)^2} \tag{C.64}$$

Recall that $w_k = \frac{a_k^2 q}{1 + a_k^2 q}$, which implies the variance factor is $v_k = w_k(1 - w_k) = \frac{a_k^2 q}{(1 + a_k^2 q)^2}$. Dividing by $q$, we map the block trace exactly to the variance factor:

$$\lim_{z \to 0^+} z^2 \Psi_k(z) = \frac{\mu_k v_k}{q} \tag{C.65}$$

Second, we use (C.61) to evaluate $T_{12}$ given by (C.60). Distributing the $z^2$ multiplier from the projection limit alongside the $v(z)^2/z^2 \to q^2$ convergence yields:

$$\lim_{z \to 0^+} \frac{1}{\beta} \left( \frac{v(z)^2}{z^2} \right) \left[ z^2 \Psi_1(z) \right] \left[ z^2 \Psi_2(z) \right] = \frac{1}{\beta} (q^2) \left( \frac{\mu_1 v_1}{q} \right) \left( \frac{\mu_2 v_2}{q} \right) = \frac{\mu_1 \mu_2 v_1 v_2}{\beta} \tag{C.66}$$

Third, we evaluate the denominator $\Delta(0)$ as $z \to 0^+$:

$$\Delta(0) = \lim_{z \to 0^+} \left[ 1 - \frac{1}{\beta} \left( \frac{v(z)^2}{z^2} \right) \sum_{k=1}^{2} \frac{\mu_k a_k^4}{(1 + a_k^2 q)^2} \right] = 1 - \frac{1}{\beta} \sum_{k=1}^{2} \mu_k \frac{a_k^4 q^2}{(1 + a_k^2 q)^2} \tag{C.67}$$

Recognizing the squared weight $w_k^2 = \left( \frac{a_k^2 q}{1 + a_k^2 q} \right)^2$, we get $\Delta(0) = 1 - \frac{1}{\beta} \sum_{k=1}^{2} \mu_k w_k^2$. We apply the identity $\beta = \sum_{k=1}^{2} \mu_k w_k$, given by (C.56), to replace the leading 1:

$$\Delta(0) = \frac{\sum_{k=1}^{2} \mu_k w_k - \sum_{k=1}^{2} \mu_k w_k^2}{\beta} = \frac{1}{\beta} \sum_{k=1}^{2} \mu_k w_k (1 - w_k) = \frac{\mu_1 v_1 + \mu_2 v_2}{\beta} \tag{C.68}$$

Finally, taking the ratio of the evaluated numerator and denominator, we get

$$T_{12} = \frac{\frac{\mu_1 \mu_2 v_1 v_2}{\beta}}{\frac{\mu_1 v_1 + \mu_2 v_2}{\beta}} = \frac{\mu_1 \mu_2 v_1 v_2}{\mu_1 v_1 + \mu_2 v_2} \tag{C.69}$$

which concludes the proof.

### C.2.3   PROOF OF LEMMA C.3

We evaluate the cross-trace by introducing a continuous, deterministic perturbation $t$ to the resolvent. We define the perturbed resolvent matrix as $R(z,t) = (A \Omega \Omega^T A + t D_2 + z I_d)^{-1}$. Let $m_1(z,t) = \frac{1}{d} \text{tr}(D_1 R(z,t))$ be its normalized trace on the first subspace.

Taking the derivative of the random trace $m_1(z,t)$ with respect to the perturbation $t$ at $t = 0$ directly yields the target cross-trace. Using the matrix derivative identity $\frac{\partial}{\partial t} M^{-1} = -M^{-1} \frac{\partial M}{\partial t} M^{-1}$:

$$\frac{\partial}{\partial t} m_1(z,t) \Big|_{t=0} = -\frac{1}{d} \text{tr} \left( D_1 R(z,0) D_2 R(z,0) \right) = -\frac{1}{d} \text{tr} \left( D_1 R(z) D_2 R(z) \right) \tag{C.70}$$

By the Bai-Silverstein theorem, $R(z,t)$ is asymptotically equivalent to the perturbed deterministic matrix $T(z,t)$. Because the perturbation $t D_2$ simply shifts the diagonal, the perturbed Stieltjes root $v(z,t)$ enforces the following exact structural form for the deterministic equivalent:

$$T(z,t) = (z I_d + v(z,t) A^2 + t D_2)^{-1} \tag{C.71}$$

Taking the derivative of the deterministic trace $\bar{m}_1(z,t) = \frac{1}{d} \text{tr}(D_1 T(z,t))$ at $t = 0$ gives:

$$\bar{m}_1'(0) = -\frac{1}{d} \text{tr} \left( D_1 T(z) \left[ v'(0) A^2 + D_2 \right] T(z) \right) \tag{C.72}$$

Because $D_1$ and $D_2$ are strictly orthogonal ($D_1 D_2 = 0$) and $T(z)$ is diagonal, the terms commute and the $D_2$ cross-term becomes zero ($D_1 T(z) D_2 T(z) = 0$). Therefore,

$$\bar{m}_1'(0) = -v'(0) \left[ \frac{1}{d} \text{tr}(D_1 A^2 T(z)^2) \right] = -v'(0) \Psi_1(z) \tag{C.73}$$

To evaluate the scalar derivative $v'(0)$, we must construct the fixed-point equation for the perturbed root $v(z,t)$. Also by the Silverstein equation (Silverstein, 1995), we have

$$\frac{1}{v(z,t)} = z + \frac{1}{\beta d} \text{tr} \left( A^2 T(z,t) \right) \tag{C.74}$$

We differentiate both sides of this fixed-point equation with respect to $t$ at $t = 0$:

$$-\frac{v'(0)}{v(z)^2} = \frac{1}{\beta d}\mathrm{tr}\left(A^2\frac{\partial}{\partial t}T(z,t)\Big|_{t=0}\right) = -\frac{1}{\beta d}\mathrm{tr}\left(A^2 T(z)\left[v'(0)A^2 + D_2\right]T(z)\right) \tag{C.75}$$

Distributing the trace operator linearly across the sum yields:

$$-\frac{v'(0)}{v(z)^2} = -\frac{v'(0)}{\beta d}\mathrm{tr}\left(A^4 T(z)^2\right) - \frac{1}{\beta d}\mathrm{tr}\left(D_2 A^2 T(z)^2\right) \tag{C.76}$$

Multiplying both sides by $-v(z)^2$ and substituting the definition $\Psi_2(z) = \frac{1}{d}\mathrm{tr}(D_2 A^2 T(z)^2)$ gives:

$$v'(0) = v'(0)\frac{v(z)^2}{\beta d}\mathrm{tr}\left(A^4 T(z)^2\right) + \frac{v(z)^2}{\beta}\Psi_2(z) \tag{C.77}$$

Grouping the $v'(0)$ terms on the left side exposes the exact macroscopic fluctuation denominator $\Delta(z)$:

$$v'(0)\underbrace{\left[1 - \frac{v(z)^2}{\beta d}\mathrm{tr}\left(A^4 T(z)^2\right)\right]}_{\Delta(z)} = \frac{v(z)^2}{\beta}\Psi_2(z) \implies v'(0) = \frac{v(z)^2}{\beta}\frac{\Psi_2(z)}{\Delta(z)} \tag{C.78}$$

Because the asymptotic limit of the random trace derivative (C.70) equals the deterministic trace derivative (C.73), we substitute the analytical solution for $v'(0)$ into the equivalence $-\lim\frac{1}{d}\mathrm{tr}(D_1 R D_2 R) = -v'(0)\Psi_1(z)$. The negative signs cancel, yielding the exact closed-form limit:

$$\lim_{d\to\infty}\frac{1}{d}\mathrm{tr}\left(D_1 R(z)D_2 R(z)\right) = \frac{v(z)^2}{\beta}\frac{\Psi_1(z)\Psi_2(z)}{\Delta(z)} \tag{C.79}$$

which concludes the proof.

## D  Proof of Proposition B.2

$(i)$ As $\beta \to 1^+$, the terms $\mathcal{T}_{inv}$ and $\mathcal{T}_{inv,\Sigma}$ diverge clearly due to the term $1/(\beta - 1)$ in (C.28), (C.41). Also as $\beta \to 1^-$, then $q \to \infty$ and so $\mathcal{T}_{inv}$ and $\mathcal{T}_{inv,\Sigma}$ diverge. This shows that $\lim_{\beta\to 1} F(\beta) = \infty$.

We next note that for $\beta \geq 1$, the model definitions dictate that $w_k = 1$, $v_k = 0$, and $T_{12} = 0$. We substitute these constants into the components of $F(\beta)$:

$$\mathrm{Bias}(\beta) = \sum_{k=1}^{2}\mu_k(\alpha_k + \delta_k)^2$$

$$\mathcal{T}_{var}(\beta) = \sum_{k=1}^{2}\mu_k s_k(g_k + \tilde{\delta}_k)^2$$

$$\mathcal{T}_{var,\Sigma}(\beta) = \sum_{k=1}^{2}\mu_k \Sigma_k(g_k + \tilde{\delta}_k)^2$$

$$\mathcal{T}_{inv}(\beta) = \frac{1}{\beta - 1}\sum_{k=1}^{2}\mu_k\frac{\Sigma_k^2}{a_k^2}$$

$$\mathcal{T}_{inv,\Sigma}(\beta) = \frac{1}{\beta - 1}\sum_{k=1}^{2}\mu_k\frac{\Sigma_k}{a_k^2}$$

By substituting these components into the objective function $F(\beta)$, we can write it in the form:

$$F(\beta) = C_1 + \frac{C_2}{\beta - 1}$$

where $C_1$ and $C_2$ are finite, strictly positive constants independent of $\beta$. The derivative is $F'(\beta) = -\frac{C_2}{(\beta-1)^2}$. Since $C_2 > 0$, we have $F'(\beta) < 0$, proving $F(\beta)$ is strictly decreasing for $\beta > 1$. As $\beta \to \infty$, the term $\frac{C_2}{\beta-1} \to 0$. The function converges to the constant $C_1$ given by: $F(\uparrow\infty)$:

$$F(\uparrow\infty) = \sum_{k=1}^{2} \mu_k (\alpha_k + \delta_k)^2 + \gamma\bar{\Sigma} \sum_{k=1}^{2} \mu_k \Sigma_k (g_k + \tilde{\delta}_k)^2$$

$(ii)$ At $\beta = 0$, the implicit variable $q = 0$, which implies $w_k = 0$, $v_k = 0$, and $T_{12} = 0$. Furthermore, the leading $q$ multiplier in $\mathcal{T}_{inv}$ and $\mathcal{T}_{inv,\Sigma}$ sets both inverse trace terms exactly to zero. Substituting these into $F(\beta)$ eliminates all cross-terms, yielding:

$$F(0) = \sum_{k=1}^{2} \mu_k \alpha_k^2 + \gamma\bar{\Sigma} \sum_{k=1}^{2} \mu_k \Sigma_k g_k^2$$

Next, we evaluate the gap $\Delta F = F(\uparrow\infty) - F(0)$:

$$\Delta F = \sum_{k=1}^{2} \mu_k \left[ (\alpha_k + \delta_k)^2 - \alpha_k^2 \right] + \gamma\bar{\Sigma} \sum_{k=1}^{2} \mu_k \Sigma_k \left[ (g_k + \tilde{\delta}_k)^2 - g_k^2 \right]$$

Based on the parameter definitions, $a_2 = r$ and $\delta_2 = \tilde{\delta}_2 = \frac{1}{1+\kappa} - \frac{\eta}{r}$. Expanding the squared perturbations for $k = 2$ yields:

$$(\alpha_2 + \delta_2)^2 - \alpha_2^2 = \left(1 - \frac{\eta}{r}\right)^2 - \alpha_2^2 = \frac{\eta^2}{r^2} - \frac{2\eta}{r} + 1 - \alpha_2^2$$

$$(g_2 + \tilde{\delta}_2)^2 - g_2^2 = \left(-\frac{\eta}{r}\right)^2 - g_2^2 = \frac{\eta^2}{r^2} - g_2^2$$

Substituting these into $\Delta F$, the leading-order behavior as $r \to 0^+$ is dominated by the $1/r^2$ terms:

$$\Delta F = \frac{\eta^2}{r^2} \mu_2 \left(1 + \gamma\bar{\Sigma}\right) + \mathcal{O}\left(\frac{1}{r}\right)$$

Because $\mu_2(1 + \gamma\bar{\Sigma})\eta^2 > 0$, the gap diverges to positive infinity as $r \to 0^+$. Thus, there exists a sufficiently small $r > 0$ such that $\Delta F > 0$, or $F(\uparrow\infty) > F(0)$.

$(iii)$ We next calculate $F'(0) = \frac{dF}{d\beta}\big|_{\beta=0}$. The asymptotic test error in the $\beta < 1$ regime is given by:

$$F(\beta) = \text{Bias} + \gamma\mathcal{T}_{inv}\mathcal{T}_{var} + \gamma\bar{\Sigma}\mathcal{T}_{var,\Sigma} + \gamma^2\bar{\Sigma}\mathcal{T}_{inv,\Sigma}\mathcal{T}_{var}$$

By applying the product rule with respect to $\beta$, the full derivative is:

$$F'(\beta) = \text{Bias}' + \gamma\left(\mathcal{T}'_{inv}\mathcal{T}_{var} + \mathcal{T}_{inv}\mathcal{T}'_{var}\right) + \gamma\bar{\Sigma}\mathcal{T}'_{var,\Sigma} + \gamma^2\bar{\Sigma}\left(\mathcal{T}'_{inv,\Sigma}\mathcal{T}_{var} + \mathcal{T}_{inv,\Sigma}\mathcal{T}'_{var}\right)$$

To evaluate this at $\beta = 0$, we must look at the inverse trace terms. Both $\mathcal{T}_{inv}$ and $\mathcal{T}_{inv,\Sigma}$ are defined with a leading factor of $q$. When $\beta \to 0$, the implicit root $q \to 0$. Because the fraction following $q$ converges to a finite constant as $q \to 0$, we have exactly:

$$\mathcal{T}_{inv}(0) = 0 \quad \text{and} \quad \mathcal{T}_{inv,\Sigma}(0) = 0$$

Substituting these zeros into the product rule eliminates the $\mathcal{T}'_{var}(0)$ terms entirely. The derivative simplifies to:

$$F'(0) = \text{Bias}'(0) + \gamma\mathcal{T}'_{inv}(0)\mathcal{T}_{var}(0) + \gamma\bar{\Sigma}\mathcal{T}'_{var,\Sigma}(0) + \gamma^2\bar{\Sigma}\mathcal{T}'_{inv,\Sigma}(0)\mathcal{T}_{var}(0)$$

We define the rightmost terms collectively as the variance penalty $V(\gamma)$:

$$V(\gamma) := \gamma\left(\mathcal{T}'_{inv}(0)\mathcal{T}_{var}(0) + \bar{\Sigma}\mathcal{T}'_{var,\Sigma}(0)\right) + \gamma^2\left(\bar{\Sigma}\mathcal{T}'_{inv,\Sigma}(0)\mathcal{T}_{var}(0)\right)$$

Hence, $F'(0) = \text{Bias}'(0) + V(\gamma)$. Because all the terms in $V(\gamma)$ are finite for any strictly positive $r > 0$, the derivatives evaluated at $\beta = 0$ are all finite constants. Since every term in $V(\gamma)$ is scaled by either $\gamma$ or $\gamma^2$, we have:

$$\lim_{\gamma \to 0} V(\gamma) = 0$$

We next derive $\text{Bias}'(0)$. Recall the definition of Bias given by:

$$\text{Bias}(\beta) = \sum_{k=1}^{2} \mu_k \left[ \alpha_k^2(1 - w_k) + (\alpha_k + \delta_k)^2 w_k \right] + T_{12}(\tilde{\delta}_2^2 - \tilde{\delta}_1^2)(\Sigma_1^2 - \Sigma_2^2)$$

By expanding the inner bracket and grouping the $w_k$ terms, we obtain:

$$\alpha_k^2 - \alpha_k^2 w_k + (\alpha_k^2 + 2\alpha_k \delta_k + \delta_k^2)w_k = \alpha_k^2 + w_k(2\alpha_k \delta_k + \delta_k^2)$$

This gives the reformulated Bias equation:

$$\text{Bias}(\beta) = \sum_{k=1}^{2} \mu_k \alpha_k^2 + \sum_{k=1}^{2} \mu_k w_k(2\alpha_k \delta_k + \delta_k^2) + T_{12}(\tilde{\delta}_2^2 - \tilde{\delta}_1^2)(\Sigma_1^2 - \Sigma_2^2)$$

To differentiate this with respect to $\beta$, we apply the chain rule via the implicit variable $q$. First, define the constant $c = \mu_1 a_1^2 + \mu_2 a_2^2$. From the defining equation $\beta(q) = \sum_{k=1}^{2} \mu_k \frac{a_k^2 q}{1 + a_k^2 q}$, we take the derivative with respect to $q$:

$$\frac{d\beta}{dq} = \sum_{k=1}^{2} \mu_k \frac{a_k^2}{(1 + a_k^2 q)^2}$$

Evaluating at $q = 0$ gives $\frac{d\beta}{dq}\big|_0 = \mu_1 a_1^2 + \mu_2 a_2^2 = c$. By the inverse function theorem, $q'(0) = \frac{dq}{d\beta}\big|_0 = \frac{1}{c}$. Now we sequentially compute the initial derivatives of the sub-components $w_k$ and $T_{12}$:

Since $w_k = \frac{a_k^2 q}{1 + a_k^2 q}$, the chain rule yields $w_k'(0) = a_k^2 q'(0) = \frac{a_k^2}{c}$. In addition, for small $q$, the variables $v_k = w_k(1 - w_k)$ expand to first order as $v_k = a_k^2 q + \mathcal{O}(q^2)$. Substituting this into the definition of $T_{12}$ gives:

$$T_{12}(q) = \frac{\mu_1 \mu_2 (a_1^2 q)(a_2^2 q)}{\mu_1(a_1^2 q) + \mu_2(a_2^2 q)} + \mathcal{O}(q^2) = q\frac{\mu_1 \mu_2 a_1^2 a_2^2}{c} + \mathcal{O}(q^2)$$

Taking the derivative with respect to $\beta$ evaluates to $T_{12}'(0) = q'(0)\frac{\mu_1 \mu_2 a_1^2 a_2^2}{c} = \frac{\mu_1 \mu_2 a_1^2 a_2^2}{c^2}$. Finally, we substitute $w_k'(0)$ and $T_{12}'(0)$ directly into the differentiated Bias equation:

$$\text{Bias}'(0) = \sum_{k=1}^{2} \mu_k w_k'(0)(2\alpha_k \delta_k + \delta_k^2) + T_{12}'(0)(\tilde{\delta}_2^2 - \tilde{\delta}_1^2)(\Sigma_1^2 - \Sigma_2^2)$$

$$= \frac{1}{c}\sum_{k=1}^{2} \mu_k a_k^2(2\alpha_k \delta_k + \delta_k^2) + \frac{\mu_1 \mu_2 a_1^2 a_2^2}{c^2}(\tilde{\delta}_2^2 - \tilde{\delta}_1^2)(\Sigma_1^2 - \Sigma_2^2)$$

For the first class ($k = 1$), the definitions give $\delta_1 = -\alpha_1$, resulting in $2\alpha_1 \delta_1 + \delta_1^2 = -\alpha_1^2$. For the second class ($k = 2$), as $r \to 0^+$, the term $a_2^2(2\alpha_2 \delta_2 + \delta_2^2) \to r^2(\eta^2/r^2) = \eta^2$. The cross-term converges to $\frac{\mu_2}{\mu_1 a_1^2}\eta^2((\rho + 1)^2 - 1)$. Summing these asymptotic components gives:

$$\lim_{r \to 0^+} \text{Bias}'(0) = \frac{1}{\mu_1 a_1^2}\left[\mu_1 a_1^2(-\alpha_1^2) + \mu_2 \eta^2\right] + \frac{\mu_2 \eta^2}{\mu_1 a_1^2}\left((\rho + 1)^2 - 1\right)$$

$$= -\alpha_1^2 + \frac{\mu_2 \eta^2}{\mu_1 a_1^2}(\rho + 1)^2$$

Since $a_1^2 = \eta^2(\rho + 1)^2$, this simplifies to:

$$\lim_{r \to 0^+} \text{Bias}'(0) = -\alpha_1^2 + \frac{\mu_2}{\mu_1}$$

Recall that $\alpha_1 = (\kappa - 1)/(\rho + \kappa)$ and $\kappa = \gamma(\mu\rho + 1 - \mu)$. Hence $\frac{d}{d\gamma}\alpha_1^2(\gamma)\big|_0 < 0$. Also,

$$\alpha_1^2(0) = \frac{1}{\rho^2} \geq \frac{1 - \mu_1}{\mu_1} = \frac{\mu_2}{\mu_1},$$

by our assumption. By continuity, for small enough $\gamma$, we have $\alpha_1^2 \geq \mu_2/\mu_1$ and so we have $\lim_{r \to 0^+} \text{Bias}'(0) < 0$. Because $\text{Bias}'(0)$ is strictly negative for small $r$, and the variance penalty $V(\gamma)$ can be made arbitrarily small for small $\gamma$, there must exist constants $r_0 > 0$ and $\gamma_0 > 0$ such that for all $r < r_0$ and $\gamma < \gamma_0$, we have $F'(0) < 0$. This completes the proof of the proposition.

# E  Gradient and Hessian Calculations for Outcome Supervision (OS) Loss

For clarity, let $M = I + VS$ (dropping the index $\tau$ for a single batch) and define the loss as $f(V) = \frac{1}{2}\|M^k w^*\|^2$.

We use the differential approach. Let $dV$ be a small perturbation in $V$. Then $dM = (dV)S$. The differential of the loss is:

$$df = \langle M^k w^*, d(M^k w^*)\rangle = (w^*)^T (M^k)^T d(M^k) w^*$$

Using the power rule for differentials, $d(M^k) = \sum_{j=0}^{k-1} M^j (dM) M^{k-1-j}$. Substituting $dM = (dV)S$:

$$df = \sum_{j=0}^{k-1} (w^*)^T (M^k)^T M^j (dV) S M^{k-1-j} w^*$$

Using the property $\text{tr}(A^T BC) = \text{tr}(CA^T B)$, we isolate $dV$:

$$df = \text{tr}\left(dV \sum_{j=0}^{k-1} SM^{k-1-j} w^* (w^*)^T (M^k)^T M^j\right)$$

The gradient $\nabla_V \mathcal{L}$ is the transpose of the matrix multiplying $dV$:

$$\nabla_V \mathcal{L} = \sum_{j=0}^{k-1} (M^T)^j M^k w^* (w^*)^T (M^T)^{k-1-j} S^T$$

We next proceed to calculate the Hessian of the loss. The gradient can be viewed as a product of terms: $G(V) = \sum_{j=0}^{k-1} A_j(V) M^k(V) B_j(V)$, with $A_j(V) = (M^T)^j$ and $B_j(V) = w^*(w^*)^T (M^T)^{k-1-j} S^T$. Applying the product rule for the differential $dG$:

$$dG = \sum_{j=0}^{k-1} \left((dA_j) M^k B_j + A_j (dM^k) B_j + A_j M^k (dB_j)\right)$$

Near the global minimum, the term $M^k w^* \approx 0$. In this regime, terms containing $M^k$ (the outer factors) vanish, leaving only the term where the differential acts directly on $M^k$. Thus,

$$dG \approx \sum_{j=0}^{k-1} (M^T)^j (dM^k) w^* (w^*)^T (M^T)^{k-1-j} S^T$$

Substituting for $d(M^k) = \sum_{j=0}^{k-1} M^j (dM) M^{k-1-j}$, we get

$$H[dM] \approx \sum_{j=0}^{k-1}\sum_{l=0}^{k-1} (M^T)^j \left(M^l (dM) M^{k-1-l}\right) w^* (w^*)^T (M^T)^{k-1-j} S^T$$

where for a direction $E$, we have $H[E] = \frac{d}{dt}\nabla_V \mathcal{L}(V + tE)\big|_{t=0}$.

We next upper bound the spectral norm of the Hessian as

$$\|H\|_{\text{op}} \leq \sum_{j=0}^{k-1}\sum_{l=0}^{k-1} \left\|M^j\right\|_{\text{op}} \left\|M^l\right\|_{\text{op}} \left\|M^{k-1-l}\right\|_{\text{op}} \left\|M^{k-1-j}\right\|_{\text{op}} \|w^*\|_{\ell_2}^2 \|S\|_{\text{op}}$$

$$\leq \sum_{j=0}^{k-1}\sum_{l=0}^{k-1} \|M\|_{\text{op}}^j \|M\|_{\text{op}}^l \|M\|_{\text{op}}^{k-1-l} \|M\|_{\text{op}}^{k-1-j} \|w^*\|^2 \|S\|_{\text{op}}$$

$$= k^2 \rho(M)^{2k-2} \|w^*\|_{\ell_2}^2 \|S\|_{\text{op}},$$

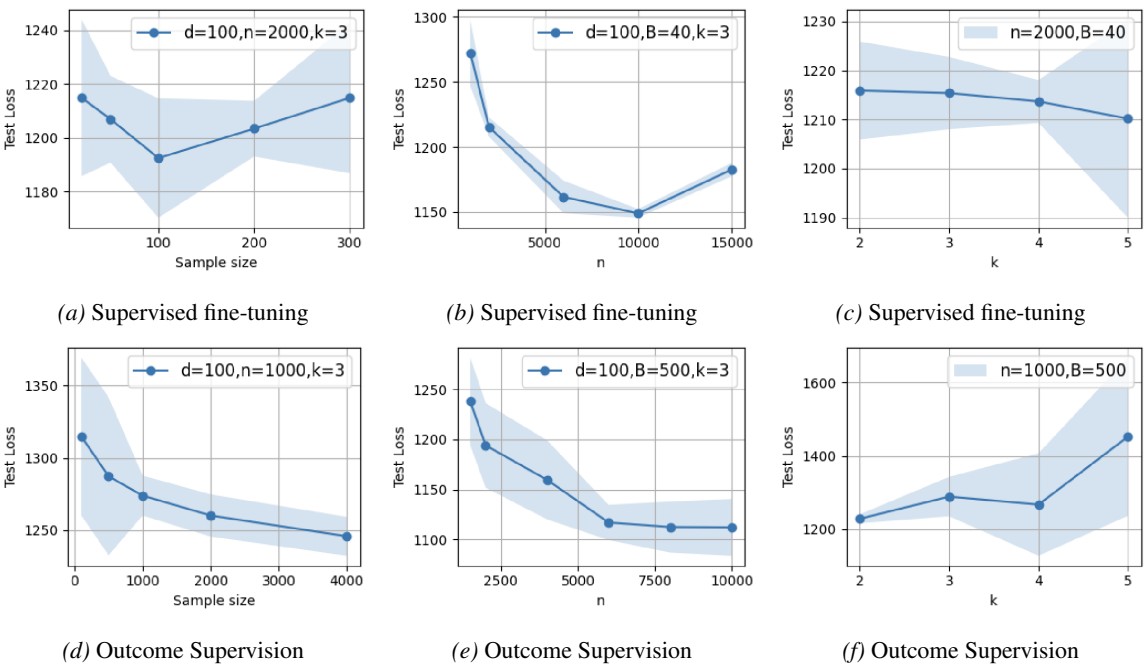

*Figure 6.* LSA experiments: Test loss for (a)-(c) post-training with SFT, and (d)-(f) post-training with Outcome Supervision (OS). For SFT, there is a turning point where larger sample size ($B$) and context-length ($n$) hurt the performance. In contrast, for OS larger $B, n$ improves the performance.

where the second step follows from sub-multiplicativity of the operator norm.

# F    Additional Experiments

## F.1    Linear Self-Attention

In this subsection, we present our results on transformers with a single linear self-attention (LSA) layer. The data setup and the pretraining, post-training, and test pipelines are the same as in Section 8.

We choose the token dimensions $d = 100$, and post-train the model for 130 epochs using Adam with learning rate $\eta = 0.001$. During inference, we return the final predicted weight vector without CoT, i.e. at test time we use $k = 1$.

Fig. 6 (a)-(c) show the results when post-training is done with the SFT loss. Fig. 6a, 6b show that increasing the sample size ($B$) or context length ($n$) initially yields a lower test loss but further increasing the sample size or context length increases the test loss. Fig. 6c shows that the test loss is relatively robust and not sensitive to the length of post-training CoT ($k$). Fig 6 (d)-(f) show the results when post-training is done with the OS loss. In contrast to SFT, Fig. 6d, 6e show that OS benefits from larger sample size ($B$) and context length ($n$), and Fig. 6f shows that longer CoT ($k$) during post-training increases the test loss and degrades the performance.

## F.2    Evaluation on Real Reasoning Benchmarks

We conduct experiments fine-tuning Qwen2.5-7B-Instruct on M23k dataset (Huang et al., 2025b), a medical reasoning dataset of 23,493 multiple-choice questions. To ensure high data quality, the dataset underwent strict decontamination and deduplication. We removed "easy" questions (those already answered correctly by Qwen2.5-7B-Instruct or Qwen2.5-32B-Instruct). Reasoning traces were distilled from DeepSeek-R1 (Guo et al., 2025), and we strictly filtered out traces that led to incorrect final answers. For SFT, we maintained the hyperparameters from (Huang et al., 2025b) (batch size 16, 5 epochs). We report average accuracy across 3 in-distribution and 7 out-of-distribution test sets. We investigate how different subset selection methods impact SFT performance. We compare (1) Random Selection as our baseline with High-Quality / Hard & Diverse: Subsets labeled by an external LLM (Gemini) for high semantic difficulty and diversity, representing the tail-end of the reasoning distribution. Around 2k examples in the data are labeled as most difficult (difficulty 5/5) by Gemini. As shown

in the table below, **high-Quality SFT data yields the best performance**, significantly outperforming random selection at much smaller data scales (e.g., 1k High-Quality matches the performance of ~3k Random). Notably, selecting even a larger number of high-quality (hard) examples results in a slight drop in performance (-0.55%) compared to that of 1k high-quality data. This is consistent with our Figure 4a and Figure 6a further confirming the conclusions of our theoretical analysis.

| Method | size | Acc (avg of 10 tasks) |
|---|---|---|
| Random | 500 | 54.41% |
| Random | 750 | 55.09% |
| Random | 1k | 55.75% |
| Random | 2k | 55.98% |
| Random | 3k | 56.22% |
| High-quality (hard and diverse) | 500 | 55.67% |
| High-quality (hard and diverse) | 750 | 56.52% |
| High-quality (hard and diverse) | 1k | **57.77**% |
| High-quality (hard and diverse) | 2k | 57.22% (-0.55%) |

*Table 1.* Accuracy comparison of Random vs. High-quality methods across different sizes.

