# OpenReview forum: "Theoretical Perspectives on Data Quality and Synergistic Effects in Pre- and Post-Training Reasoning Models"
_ICML.cc/2026/Conference — ICML 2026 regular_

### Official Review · Reviewer_w29J · 2026-03-02

**Soundness:** 3
**Presentation:** 2
**Significance:** 3
**Originality:** 2
**Overall Recommendation:** 3
**Confidence:** 3

**Summary:**

This paper presents a theoretical framework to analyze data strategies across the three major stages of large language model training: pre-training, supervised fine-tuning (SFT), and reinforcement learning (RL). Using a carefully designed synthetic dataset, the authors investigate how different data characteristics affect capability development at each stage.

The key insights can be summarized as follows:

1. Balanced data during pre-training encourages the emergence of latent capabilities.

2. SFT benefits from a relatively small but high-quality dataset.

3. RL requires large-scale data that is well aligned with the capabilities developed during pre-training.

These findings are supported by both theoretical analysis and empirical validation on synthetic experiments.

**Compliance With Llm Reviewing Policy:**

Affirmed.

**Final Justification:**

The rebuttal has addressed most of my concerns and significantly improved my confidence in the paper. I am now leaning toward a weak accept.

**Key Questions For Authors:**

As stated in weaknesses, my main concerns are about (1) the limited experiments and (2) the novelty of the insights. The empirical validation is conducted only on synthetic datasets with GPT-2-scale models, and it is unclear whether the theoretical conclusions would hold in realistic large-scale LLM settings. In addition, many of the findings appear consistent with existing empirical observations, and it would be helpful to better clarify what fundamentally new insights or predictive power the theory provides beyond confirming prior intuition. If the authors can provide stronger justification for practical scalability and more clearly discuss the novel contributions of the framework, I would be open to increasing my score.

**Limitations:**

I would encourage the authors to explicitly include a limitation section discussing the experimental constraints of the work.

**Strengths And Weaknesses:**

Strengths

1. Data selection and data strategy are central to modern LLM training pipelines, spanning pre-training, SFT, and RL. Providing a unified theoretical perspective on these stages is valuable for the community.

2. The paper develops a rigorous theoretical framework on synthetic data, offering formal explanations for several empirical observations reported in prior work. This helps improve our understanding of how data distribution influences capability formation.

3. The paper is well organized, and the progression from problem setups to theoretical results and then to empirical validation is generally easy to follow.


Weaknesses

1. The empirical validation is conducted primarily on synthetic datasets using GPT-2-scale models. While GPT-2 is large in controlled experiments, the gap between synthetic setups and real-world LLM training is substantial. It remains unclear whether the theoretical conclusions hold under realistic data distributions and modern large-scale models. Stronger validation on more practical settings would significantly strengthen the paper.

2. Many of the high-level conclusions, such as the importance of balanced pre-training data, high-quality SFT data, and alignment between RL data and pre-training capabilities, have already been observed empirically in prior works. While the theoretical formalization is valuable, the paper would benefit from deeper or more surprising insights that go beyond confirming existing empirical intuitions. Providing more principled guidelines or uncovering non-obvious trade-offs would improve the contribution.

3. The theoretical sections are mathematically dense, which may limit accessibility. Additional intuitive explanations, visualizations, or empirical case studies would help readers better understand the practical implications of the theory. Currently, the presentation leans heavily toward formal derivations, with comparatively limited discussion.

4. The paper equates RL with outcome-based rewards. However, recent work has explored process rewards and intermediate supervision in RL-based LLM training. A discussion comparing outcome rewards and process rewards within the proposed theoretical framework would make the analysis more complete.

---

> ### Author Rebuttal · Authors · 2026-03-30
>
> Thanks for your review! You raised great points and we respond to your questions below.
>
> **Response to Weaknesses**
>
> **W1.** While our theoretical analysis follows the LSA and linear regression setting studied in prior work (Huang et al., 2025b; Javanmard et al., 2025), we have confirmed the validity of our conclusions on GPT-2 from random initialization. For language tasks, as discussed in the introduction and related work sections, there is recent ample evidence for our theoretical results confirming our results: (insight 1) hard examples are best for SFT (Muennighoff et al., 2025; Guha et al., 2025; Huang et al., 2025b, Akter et al., 2025); (insight 2) SFT data should be small and high-quality, i.e. hard and diverse (Muennighoff et al., 2025; Guha et al., 2025; Huang et al., 2025b, Akter et al., 2025); (insight 3) RL data should be large (OpenAI o1, DeepSeek R1); (insight 4) pretraining data should be diverse (Akter et al., 2025). We view additional language experiments as adding limited new evidence beyond the strong existing literature, though we included new experiments below to strengthen empirical validation.
>
> **W2.** Great comment. Indeed, while our analysis mainly shed light onto several recent empirical observations, they also provide several new insights:
>
> **SFT.** Recent empirical results quantified high-quality SFT data as a small subset of hard and diverse examples. Our analysis provides further insights: if the SFT data is not high-quality (hard) enough, then increasing its size yields better performance. Theoretical insights behind this conclusion is discussed in our response to reviewer Dv7P, see W3, Q1,Q2. (Large overlap corresponds to examples that are not hard enough for the pretrained model, so it is better to increase B; see also the new experiment on Qwen 2.5 below)
>
> **RL.** While recent empirical work (e.g. *Does Reinforcement Learning Really Incentivize Reasoning Capacity in LLMs Beyond the Base Model?*, NeurIPS’25 oral) concluded that RL cannot learn new skills beyond what’s known by the pretrained model, our analysis reveal that this is indeed correct, **if post-training is done on relatively small data**. On the other hand, having access to a very large pool of data, RL can learn new skills; See Insight 3 where we discuss that large training data can mitigate instability of RL in learning new tasks. Our finding explains the success of OpenAI-o1 and DeepSeek-R1 reasoning models that are trained by RL on very large datasets.
>
> **W3.** Thanks for pointing out, we will improve the presentation in our revision (using the extra page in camera ready).
>
> **W4.** This is a great suggestion. Mathematically, process rewards are equivalent to our Supervised Fine-Tuning (SFT) formulation operating under **partial supervision**. While standard SFT provides dense step-by-step targets, Process Reward Models typically evaluate specific intermediate reasoning milestones (as in practice, process rewards are rarely dense at the token level and annotators typically provide rewards only at discrete milestones, e.g., at the end of a sentence or a specific mathematical derivation). We can formalize this by defining a mask $M \subseteq \{1, \dots, k\}$ representing the specific steps that receive a process reward. This smoothly bridges the paradigms:(1) $M = \{1, \dots, k\}$ (full SFT loss) (2) $M = \{k\}$ (outcome supervision) (3) $1 < |M| < k$ (Process reward model)
>
> Limitations. Thanks for the suggestion, we will add a limitation section.
>
> **New Experiment:**
>
> To further validate our approach, we fine‑tuned Qwen 2.5‑7B‑Instruct on M23k, a medical reasoning dataset of 23,493 multiple‑choice questions. Please refer to Huang et al. (2025) for details and the experimental setup. We investigate how different subset selection methods impact SFT performance. We compare (1) Random Selection as our baseline with (2) High-Quality / Hard & Diverse: Subsets labeled by an external LLM (Gemini) for high semantic difficulty and diversity, representing the tail-end of the reasoning distribution. Around 2k examples in the data are labeled as most difficult (difficulty 5/5) by Gemini.
> As shown in the table below, **high-Quality SFT data yields the best performance**, significantly outperforming random selection at much smaller data scales (e.g., 1k High-Quality outperforms the performance of ~3k Random). Notably, selecting even a larger number of high-quality (hard) examples results in a slight drop in performance (-0.55%) compared to that of 1k high-quality data. This is consistent with our Fig 3(a) and Fig 4(a) further confirming the conclusions of our theoretical analysis.
> | Method|Size|Acc (avg of 10 tasks)|
> |---|---|---|
> | Random|500|54.41%|
> | Random|750|55.09%|
> | Random|1k|55.75%|
> | Random|2k|55.98%|
> | Random|3k|56.22%|
> | High‑quality (hard + diverse) |500| 55.67%|
> | High‑quality (hard + diverse) |750|56.52%|
> | High‑quality (hard + diverse) |1k|**57.77%**|
> | High‑quality (hard + diverse) |2k|57.22 (–0.55%)|

---

> > ### Author Rebuttal · Reviewer_w29J · 2026-04-02
> >
> > Most of my concerns have been addressed. Consequently, I have updated my score.

---

> > > ### Author Response · Authors · 2026-04-02
> > >
> > > Thank you for the update and for acknowledging our revisions! We noticed that while the comment mentions an updated score, the system still displays a **3: Weak Reject, with no change from the previous score**. We wanted to bring this to the reviewer’s attention in case a technical glitch prevented the updated score from being saved in the portal. We would be grateful if this could be cross-checked to ensure the final evaluation accurately reflects the reviewer's current assessment.

---

### Official Review · Reviewer_XRmH · 2026-03-11

**Soundness:** 2
**Presentation:** 2
**Significance:** 3
**Originality:** 3
**Overall Recommendation:** 3
**Confidence:** 4

**Summary:**

This paper studies how pretraining and post-training data properties jointly shape model performance, using an in-context weight prediction task for linear regression. The paper focus a transformer with a single linear attention layer pretrained on data with fixed covariance, then post-trained via either Supervised Fine-Tuning (SFT) or Outcome Supervision (OS, equivalent to RL) on data with a shifted covariance. The main theoretical results include (1) SFT loss minimizer under the setting, (2) suggest that a data selection principle favoring hard examples (3) analyzes how excessive SFT data can actually degrade performance through interference (4) analyze gradient of OS landscape and related training stability. Experiments are done with the single-layer linear(SLA) and GPT-2 architecture are to validate these theoretical claims.

**Compliance With Llm Reviewing Policy:**

Affirmed.

**Final Justification:**

Rebuttal response has addressed mosf of my concerns.

**Key Questions For Authors:**

*Q1* Can you justify the CoT form of chosen Chain of Thought process of exponentially decaying errors? Shouldn’t we consider other forms of chain of thoughts such as intermediate steps from gradient-based optimization methods or else?

*Q2* Regarding the first question, I wonder how each proposed theoretical restyle ( Theroems 4.1~4.5) are affected by the choice of specific form of intermediate steps in CoT. How are the changes in the loss coupled?

*Q3*  In line 436 : Can you provide quantitative comparisons between the theoretical predictions and the GPT-2 experimental results, and explain why the k values differ between LSA experiments and the theory (k>1)? I wonder how this choices affects to the results.

*Q4* Most importantly, how much portion of these findings can be translated to the natural-language problem that LLMs are originally focusing? At least discussions and acknowledgment of limitations seems required.

**Limitations:**

The authors do acknowledge some limitations of their theoretical framework, particularly the restriction to linear regression and the single LSA layer.  However, several important limitations are not adequately discussed. The specific choice of modeling intermediate steps of chain of thoughts are not justified enough. The mismatch between theoretical assumptions and experimental parameters (the k values) is not discussed enough. The absence of any natural language experiments or larger-scale models is a significant limitation that desreves more honest engagement, especially given a title that promises insights about "Reasoning Models."

**Strengths And Weaknesses:**

*S1*. The paper tackles important questions of mechanism behind SFT and RL and how data fraction affects pretraining and post-training performance.

*S2*.  Understnading the pretraining SFT and Outcome supervision (OS) by controlling the number of CoT steps are novel approach and theoretically grounded. This enables control of complexity so that it can be promoted and widely used for future works.

*S3*.  The authors validate on two architectures (LSA and GPT-2) and provide complete proofs in the appendices.

*W1* Most of the theoretical claim largely based on LSA and linear regression weight prediction task, but never explained how sensitive are the results when architecture or tasks are changed. there are no natural language tasks. More importantly, there is no quantitative comparison against the theoretical predictions; only qualitative "agreement" is claimed.

*W2* Overlap with prior works : The interference mechanism in Insight 2, where excessive fine-tuning distorts pretrained features under distribution shift, appears closely related to Kumar et al., who prove exactly this phenomenon. The Section 7 claim about balanced diverse pretraining seems to recapitulate the diversity threshold theory of Raventos et al. These overlaps deserves explicit discussion or acknowledgement.

*W3* In the paper outcome Supervision is modeled as outcome-supervised regression, stripping away policy sampling, advantage estimation, and KL constraints, yet the paper frequently refers to "RL" interchangeably with OS. This can be misleading as  the connection to reasoning models is not clear.

References:
- Kumar, A., Raghunathan, A., Jones, R., Ma, T., and Liang, P., 2022, "Fine-Tuning can Distort Pretrained Features and Underperform Out-of-Distribution," ICLR 2022, arXiv:2202.10054
- Raventos, A., Paul, M., Chen, F., and Ganguli, S., 2023, "Pretraining Task Diversity and the Emergence of Non-Bayesian In-Context Learning for Regression," NeurIPS 2023, arXiv:2306.15063

---

> ### Author Rebuttal · Authors · 2026-03-30
>
> We thank the reviewer for valuable feedback.
>
> **Response to Weakness**
>
> **W1.** While our theoretical analysis follows the LSA and linear regression setting studied in prior work (Huang et al., 2025b; Javanmard et al., 2025), it is validated on GPT-2 from random initialization. For language tasks, as discussed in the intro, consistent results across recent studies support our main insights: (insight 1) hard examples are best for SFT (Muennighoff et al., 2025; Guha et al., 2025; Huang et al., 2025b, Akter et al., 2025); (insight 2) SFT data should be small and high-quality, i.e. hard and diverse (Muennighoff et al., 2025; Guha et al., 2025; Huang et al., 2025b, Akter et al., 2025); (insight 3) RL data should be large (OpenAI o1, DeepSeek R1); (insight 4) pretraining data should be diverse (Akter et al., 2025). We view additional language experiments as adding limited new evidence beyond the strong existing literature, though we include new experiments on Qwen 2.5 in response to reviewer w29J.
>
> **W2.** While some ideas may seem similar, there is no overlap between our results and Kumar et al; they consider contrastive pretraining on images followed by SFT or linear probing with labels. It concludes that when pretraining learns high-quality features, linear probing achieves higher OOD performance while fine-tuning achieves higher in-distribution (ID) performance. Our paper analyzes a very different setting: post-training transformer architecture on Chain of Thoughts (CoT) by supervising the intermediate steps of thinking (SFT) or supervising the outcome of CoT (OS). Importantly, our evaluation does not consider only ID or OOD, but requires doing well on both ID and OOD tasks. In addition, our insights 1-3 on requirements on SFT and OS data holds regardless of the quality of the features learned by the pretrained model.
>
> Similarly, there is no overlap between our results and Raventos et al, as they study the effect of pretraining data diversity on ICL performance. On the other hand, we argue in Sec 7 that while SFT can partially mitigate issues with imbalance pretraining, *balanced pretraining data is crucial for OS*. This setting is neither considered or studied by Raventos et al.
>
> **W3.** We have been very clear about this distinction on page 1, where we discuss OS versus RL. OS is a simpler setting—without advantage estimation or policy sampling—and therefore more suitable for theoretical analysis. However, given the fundamental similarities between OS and RL, we believe our Insight 4 extends naturally to RL variants. This is supported by recent empirical work on reasoning models (OpenAI o1, DeepSeek R1). We will further clarify these connections and contrasts in the revised version.
>
> **Response to Questions**
>
> **Q1.** We choose the exponentially decaying sequence $w_{i,\tau} = (1-(1-\eta)^i) w_\tau^\ast$ as an analytically clean proxy for gradient-based optimization.
>
> **Connection to Gradient Descent:** In *population* linear regression with identity covariance, gradient descent follows $w_i = w_{i-1} - \eta (w_{i-1} - w_\tau^\ast)$ giving $w_{i,\tau} = (1-(1-\eta)^i) w_\tau^\ast$, identical to our CoT sequence.
>
> **Tractability:** Using empirical updates $w_{i,\tau} = w_{i-1,\tau} - \frac{\eta}{n} X^\top(Xw_{i-1,\tau} - y)$ would repeatedly involve the sample covariance $S_\tau$, making analysis over multiple steps intractable. The exponential form captures the essence of iterative refinement without this complexity.
>
> **Q2.** Replacing our exponentially decaying targets with empirical gradient descent steps would change the closed-form solutions and convergence constants, though the overall framework remains intact.
>
> Theorem 4.1 (Minimizer): Our current form yields the clean solution $\tilde{V_*}= -\eta \Omega \Phi^{\dagger} -\Gamma_0^{-1}(I - \Phi \Phi^{\dagger})$. With empirical GD, the SFT targets ​$w_{i,\tau}$ become polynomials in the empirical covariance $S_\tau$, making $\tilde{V_*}$ a complex nonlinear function tied to higher-order moments of $S_\tau$.
>
> Theorem 4.2 (Convergence): The constant $c_k = \sum_{i=0}^k \rho^{2i}$ arises from geometric decay; with empirical GD, it would depend on the spectral spread of each $S_\tau$, tightening the learning-rate bound $\gamma$ based on *worst-case conditioning*.
>
> **Q3.** In our GPT-2 experiments we consider a much bigger non-linear architecture, starting from random initialization and perform CoT (k>1) at test time. This setting is much closer to training reasoning LLMs on natural language tasks. While this setting violates our theoretical assumptions, the results of our GPT-2 experiments closely follow the results of our theoretical analysis (c.f. Fig 3-GPT2 in main paper with Fig 4-LSA in Appendix). This confirms the validity of our results in more realistic scenarios. Similar results hold for k=1 at test time, but in practice as we train reasoning models to think at test time, we used k>1 for our GPT-2 experiments.
>
> **Q4.** Please see our answer to W1.

---

> > ### Author Rebuttal · Reviewer_XRmH · 2026-04-03
> >
> > The rebuttal response addressed most of my concern efficiently. I have raised my score accordingly.

---

> > > ### Author Response · Authors · 2026-04-03
> > >
> > > Thank you for the update and for acknowledging our revisions! We noticed that while the comment mentions an updated score, the system still displays a **3: Weak Reject, with no change from the previous score**. We wanted to bring this to the reviewer’s attention in case a technical glitch prevented the updated score from being saved in the portal. We would be grateful if this could be cross-checked to ensure the final evaluation accurately reflects the reviewer's current assessment.

---

### Official Review · Reviewer_dLo4 · 2026-03-12

**Soundness:** 3
**Presentation:** 3
**Significance:** 3
**Originality:** 3
**Overall Recommendation:** 4
**Confidence:** 3

**Summary:**

This work investigates the synergistic effects in pretraining, SFT, and RL (simplified as OL). Under a synthetic setting, the authors provide a theoretical analysis, including that an ideal setup for SFT is providing a small amount of high-quality data, but for RL, it is desirable to provide a large amount of data with moderate difficulty. The theoretical insights are validated by a set of controlled experiments.

**Compliance With Llm Reviewing Policy:**

Affirmed.

**Final Justification:**

The rebuttal has resolved my concerns. I maintain my positive score.

**Key Questions For Authors:**

- L273 “When interference is strong, the error remains above its value at optimal $B$ even in the large $B$ limit”: Fig.1(c) shows that this is not the case when r is even larger. I understood that the larger value of $r$ leads to stronger interference? If this is correct, how can the trend observed in Fig.1(b) and Fig.1(c) be explained?

**Limitations:**

Although the authors explicitly state that some simplifications are deliberately made for theoretical analysis, it would be helpful to include a separate paragraph discussing the remaining challenges and how the gap between theory and practice would affect practical insights.

**Strengths And Weaknesses:**

### Strengths

- Although I couldn’t verify the proof in detail, as this is not my immediate expertise, I’d say that the theoretical construction seems to be rigorous and the problem formulation is introduced in the main text clearly.
- The insights derived from a rigorous theoretical analysis can guide a better model training strategy in practice.

### Weaknesses

- Strong assumptions: Several main results rely on a highly simplified architecture (LSA), specific parameter initialization, and specific task structures. The verification of the applicability of these insights (either by theory or experiment)  after relaxing the assumptions (e.g., robustness to different parameter initializations) remains unclear.
- While the theoretical analysis provides several insights, the connection to plain-word insights is sometimes loose, and I think they need a more thorough discussion for justification.
    - Deriving Insight 2 from Sec.5.2 can be phrased more carefully, as there is nothing we can think about the ‘quality’ in the synthetic data the authors considered, where the inputs consist of randomly sampled Gaussians. It would be great to discuss more about what ‘quality’ means (e.g., how it connects to the notion of 'difficulty') and how it connects to the insights from the theoretical analysis
    - Similarly, I think there could be a more formal justification for which OS reflects the true behavior and dynamics of RL and what is missing. For example, does OS reflect the mode-seeking behavior of RL?
    - The connection to “overtinking” and the model being in the unstable region sounds a bit hand-wavy. Should ‘thinking less’, like reducing the number of demonstrations, be helpful when a model is located in such an unstable region?

---

> ### Author Rebuttal · Authors · 2026-03-30
>
> Thanks for your review and feedback! You have raised great points and we respond to your concerns and questions below.
>
> **Response to Weaknesses**
>
> **W1:** We acknowledge the simplifying assumptions in our analysis (e.g., LSA, specific initialization, and structured tasks which follow prior theoretical work on transformers, e.g. Huang et al ‘25 and Javanmard et al ‘25). That said, most prior work on pre-training/post-training synergy and SFT versus RL has been empirical, and our work seeks to provide one of the first principled theoretical accounts of these interactions. Such abstraction is necessary for rigorous analysis, and LSA provides a tractable framework for this purpose. Despite these simplifications, our high-dimensional model with low-rank post-training adaptation captures important effects, including catastrophic forgetting via interference between pre-training and post-training data. Our GPT-2 experiments (larger nonlinear model, random initialization) and recent empirical results (Muennighoff et al., 2025; Guha et al., 2025; Huang et al., 2025b, Akter et al., 2025) confirm the validity of our conclusions for training reasoning models in more realistic settings, including natural language.
>
> **W2:**  Please see our responses to each bullet point below:
>
> 1) The quality here is intended to capture new information not already learned by the pretrained model. It is therefore closely related to “hardness,” since these examples are also difficult for the pretrained model to predict.
>
> 2) Our intention is that the OS captures the key outcome-level objective of RL, not its full optimization dynamics. In particular, it reflects the fact that RL primarily favors trajectories that lead to high-reward final answers, which is consistent with the mode-seeking tendency often associated with RL. As discussed in page 1, what is missing are the algorithmic details of standard RLHF, such as sampling noise, policy-gradient updates, and advantage estimation, and as we made in clear in the paper our analysis should be viewed as a principled abstraction rather than a faithful dynamical model of full RLHF.
>
> 3) This is exactly the intuition we want to convey. In our analysis, increasing the reasoning length $k$ makes the landscape sharper near the stability boundary, so reducing $k$ can indeed help by moving the model away from this high-curvature regime and making it less sensitive to small perturbations. That said, our point is not that “thinking less” is universally better, but that when the model is already near an unstable region, shorter reasoning can be more robust by avoiding the cliff-like behavior induced by large $k$.
>
> **Response to Questions**
>
>  Regarding the effect of $r$ please see our response to Reviewer Dv7P (W3, Q1,Q2). In short, as we discussed the strength of interference is captured by $1/r$ as also appearing in our analysis.

---

> > ### Author Rebuttal · Reviewer_dLo4 · 2026-04-04
> >
> > Thank you for the response, which has resolved my concerns. I maintain my positive score.

---

### Official Review · Reviewer_Dv7P · 2026-03-13

**Soundness:** 3
**Presentation:** 2
**Significance:** 3
**Originality:** 3
**Overall Recommendation:** 4
**Confidence:** 3

**Summary:**

This paper investigates the impact of various attributes of (pre/post)training dataset on the post-training capabilities of LLMs, where the types of post-training investigated are 'process supervised' (SFT) and 'outcome supervised' (RL). The impacts they investigate are: how pretraining data balance and post-training data hardness/variety impacts the models ability to adapt to new skills while minimizing interference with capabilities acquired during pretraining. They also compare postraining data attributes for both RL and SFT settings, and compare the optimisation landscape of RL and SFT. They do this by analysing a toy model comprising of an 'in-context' weight prediction task that simulates 'chain of thought' reasoning devised and previously analysed, solved using a linear self-attention unit. They establish theorems characterising the minimizer of the problem in the SFT setting and its convergence conditions and use these to comment on the optimal data allocation for best performance in a downstream task with a modified test data distribution. They validate findings on more complex architectures.

**Compliance With Llm Reviewing Policy:**

Affirmed.

**Final Justification:**

I maintain my score supporting the paper. The authors clarified the questions I had during the rebuttal.

**Key Questions For Authors:**

- Authors mention that 'when interference is strong, the error remains above its value at optimal B even in large B limit (Fig. 1b)'. Is this true? Is the rate of convergence not just slower?
- Why does a smaller non-zero r give more interference but r= 0 gives optimal (minimal) interference? Can this be squared with insight 1, since decreasing r is increasing the post- training task alignment with the difference in post-testing and pretraining task alignment? Or are these already compatible and I'm missing something. What happens at even larger r?
- The authors use the term 'overthinking', can they define it in the context of the model?
- Would the task be able to be completed without required iterative steps? Could it be trained out the box for ICL like in other instances?
- I am struggling to intuit how to interpret the difference between the post-training and post-testing. In relating to finetuning LLMs in practice, typically for reasoning the RL process. I guess optimal A selects for have most difficult examples and a variety of them.
- Out of interest, have you considered making $\Delta$ a random variable? Or is this not tractable/ would this not qualitatively change the results?

**Limitations:**

Yes

**Strengths And Weaknesses:**

**Strengths**
- This paper presents novel work with theories that have implications of interest to the LLM interpretability community.
- I found the outcome supervised framing both an original and compelling setup for studying LLM RL post-training with a final answer that depends upon previous steps in the reasoning chain.
- There are experimental checks on a gpt-2 architecture, validating some of the findings, this is satisfying to see in a theory paper.
- The authors' work helps build rigorous understanding based on theory to several empirically observed phenomena reported in previous literature.

**Weaknesses**
- The paper would benefit from placing the existing empirical results in a more concise way, for instance, the authors mention empirical preference for small high-quality datasets in practice, but give no reference to existing literature.
- Some simulation/experimental verification for Insight 1 would be beneficial, currently there is none.
- There is possibly a contradiction in Insight 1 and the results presented in Figure 1, which I mention more in the below. If this truly is a contradiction then the result cannot be a main takeaway of the paper.
- The readability could be improved. For example, it is not clear what exactly the pretraining task is and I had to to back to the previous literature to grasp it. Also, the results and implications of the theorems could benefit from more direct comparison in the PS and OS settings. This would help with reinforcing the claims stated in the beginning of the paper: 'How does the RL optimization landscape differ from that of SFT, and when can RL achieve outcomes comparable to SFT?'. A corresponding result for data scaling in OS would be useful.
- It is confusing that, in Section 5.1 the post-testing was done on a distribution which was effectively a 'blind spot' to the model, then in Section 5.2 post-testing was done on a distribution whose covariance had identical support to the pretraining distribution. I am confused as to how this is a suitable setup for analysing pre/post interference, since the tasks are essentially identical, except for the requirement of taking the answer after 'k' steps in SFT. Then, Figure 1 caption states that the pretrained covariance has non-zero support everywhere.  This is contradictory to what is stated in the beginning of Section 5.2 (is there a typo?). I believe the correct setup may actually be the one defined in Figure 1 caption?

---

> ### Author Rebuttal · Authors · 2026-03-30
>
> Thanks for your feedback! You raised great points and we respond to your questions below.
>
> **Response to Weaknesses**
>
> **W1.** We have listed four references in lines 61-64, right column of our related work: (Muennighoff et al., 2025; Guha et al., 2025; Huang et al., 2025b), and (Akter et al., 2025). We will add these to our introduction as well, thanks for the note.
>
> **W2.** We have conducted the following experiments to verify insight 1 and will add the plots to the revision. We consider the setup in Section 5.2 with pretraining covariance $\Sigma_0 = diag(\rho 1_m, 1_{d-m})$ and post-test covariance $\Delta = diag(1_m,0_{d-m})$. As discussed in Sec 5.1 (top of page 4), the hard distributions for the pretrained model are those with covariance spanned by $range(\Delta)$. We consider data covariance $A = diag(\cos(\theta)1_m, \sin(\theta)\sqrt{(d-m){m}}1_{d-m})$, with $\theta\in (0,\pi/2)$. So smaller $\theta$ corresponds to larger alignment with the shift $\Delta$ and hence harder distribution for pretrained models.
>
> In the first experiment, we vary $B$ and report test error (5.1). As we see the min test error over B is achieved for the smallest $\theta$ and increases in $\theta$. This confirms insight 1 that harder examples are better for SFT. Also for each $\theta$ the min is achieved for a mid-value of $B$, consistent with insight 2 that SFT data should be moderately small in volume.
> |B|45|125|205|285|365|
> |-|-|-|-|-|-|
> |$\theta=0.05\pi$|1.64|1.25|**1.02**|1.17|3.03|
> |$\theta=0.15\pi$|1.69|1.39|1.16|**1.05**|1.41|
> |$\theta=0.25\pi$|1.75|1.52|1.32|**1.16**|1.32|
> |$\theta=0.35\pi$|1.82|1.67|1.49|**1.32**|1.54|
> |$\theta=0.45\pi$|1.89|1.87|1.79|**1.79**|3.42|
>
> The next is a similar experiment where we vary n (prompt length)
> |n|20|120|220|320|420|520|620|
> |-|-|-|-|-|-|-|-|
> |$\theta=0.05\pi$|1.21|1.06|0.99|0.95|0.93|**0.92**|0.93|
> |$\theta=0.15\pi$|1.10|1.00|**0.97**|0.97|1.00|1.05|1.11|
> |$\theta=0.25\pi$|1.08|1.00|**0.99**|1.04|1.11|1.21|1.31|
> |$\theta=0.35\pi$|1.10|**1.03**|1.05|1.13|1.24|1.38|1.53|
> |$\theta=0.45\pi$|1.21|**1.13**|1.19|1.32|1.49|1.68|1.89|
>
> **W3, Q1,Q2**: (effect of $r$): There is no contradiction in insight 1 and Figure1. When $r=0$, the pretrained subspace lies in the null space of the feature matrix, so the pseudo-inverse removes it—yielding no interference. For small $r>0$, weakly overlapping pretrained features cause near-zero eigenvalues in the covariance. Because the interpolating estimator must invert these near-zero eigenvalues to fit the training data, the resulting learned weights go up and this variance amplification increases the test error. Large $r>0$ ensures that the minimum eigenvalues associated with these features are well-conditioned, allowing stable inversion and effective integration of pretrained skills. This pattern aligns with our theory in the appendix B, where terms proportional to $1/r$ appear in the test error.
>
> We have also developed a new theoretical result which also resolve Q1:
>
> **(new) Theorem** Let $F(\beta)$ be the asymptotic limit of the test error derived in the proportional regime of Appendix B, viewed as a function of $\beta = B/d$. Then $F(\beta)$ is strictly decreasing for $\beta>1$. Also, as $\beta\to\infty$ it converges to a finite floor given by $F(\infty)$ (the exact expression will be given in the formal version). In addition, $F(\infty) - F(0) =C/r^2+O(1/r)$ for a fixed $C>0$. So for sufficiently small $r>0$, we have $F(\infty)> F(0)$, implying that the global minimum is strictly achieved in the low sample size regime $\beta<1$.
>
> **W4:** A direct comparison between PS and OS is difficult because defining a fair basis is unclear. PS (or SFT) uses intermediate, correlated targets via $w^*_\tau$, making the effective sample size relative to OS hard to quantify.
>
> **W5:** It is a typo. It should be $\Sigma_0=diag(\rho1_m, 1_{d-m})$.
>
> **Response to Questions**
>
> **Q1,Q2**: See response to W3.
>
> **Q3**: In our setup, the model produces a sequence of estimates for each prompt before the final output ( Eq. 3.3). Let
> k denote the chain length. Overthinking occurs when test error increases with k (as in Fig. 3f)—i.e., more thinking worsens performance.
>
> **Q4**: Setting $k=1$ disables CoT at inference, but will impact performance. In more complex cases, $k$ can vary across prompts.
>
> **Q5**: In our framework, post-training adapts the model to downstream tasks, while post-testing evaluates it on queries requiring either pretrained or newly learned skills. This parallels finetuning vs. deployment: post-training is adaptation, post-testing is evaluation. The key goal is to improve downstream performance without forgetting pretrained competencies.
>
> **Q6**: Note that $\Delta$ denotes perturbation to the “covariance” during post-testing, while the data itself remains random. We keep $\Delta$ general in our discussions—assuming only that it is low-rank—although the experiments focus on a specific choice of $\Delta$.

---

> > ### Author Rebuttal · Reviewer_Dv7P · 2026-04-03
> >
> > I thank the authors for their time and effort on the rebuttal and for answering my questions.
> > I remain positive about the work and maintain my score.

---

### Decision · Program_Chairs · 2026-04-30

**Decision:**

Accept (regular)

**Comment:**

This paper offers a study of how data properties across pretraining and post-training shape downstream capabilities, with a particular comparison between SFT and outcome-supervised/RL-style optimization.

Reviewers found the framing novel and important, especially the attempt to unify these stages within a single analytical framework and to formalize several phenomena that have previously been observed only empirically. The paper’s main strengths are its rigorous theoretical treatment, the clarity of its synthetic construction, and the presence of controlled empirical validation on both simplified and GPT-2-scale architectures.

The overall consensus is positive. I also note that Reviewers XRmH and w29J indicated in discussion that they were leaning weak accept, although their final scores were not updated in time.

I therefore recommend accept.